# Realistic Evaluation of Transductive Few-Shot Learning

**Olivier Veilleux** *
ÉTS Montreal

**Malik Boudiaf** *
ÉTS Montreal

**Pablo Piantanida**
L2S, CentraleSupélec CNRS
Université Paris-Saclay

**Ismail Ben Ayed**
ÉTS Montreal

## Abstract

Transductive inference is widely used in few-shot learning, as it leverages the statistics of the unlabeled query set of a few-shot task, typically yielding substantially better performances than its inductive counterpart. The current few-shot benchmarks use perfectly class-balanced tasks at inference. We argue that such an artificial regularity is unrealistic, as it assumes that the marginal label probability of the testing samples is known and fixed to the uniform distribution. In fact, in realistic scenarios, the unlabeled query sets come with arbitrary and unknown label marginals. We introduce and study the effect of arbitrary class distributions within the query sets of few-shot tasks at inference, removing the class-balance artefact. Specifically, we model the marginal probabilities of the classes as Dirichlet-distributed random variables, which yields a principled and realistic sampling within the simplex. This leverages the current few-shot benchmarks, building testing tasks with arbitrary class distributions. We evaluate experimentally state-of-the-art transductive methods over 3 widely used data sets, and observe, surprisingly, substantial performance drops, even below inductive methods in some cases. Furthermore, we propose a generalization of the mutual-information loss, based on $\alpha$-divergences, which can handle effectively class-distribution variations. Empirically, we show that our transductive $\alpha$-divergence optimization outperforms state-of-the-art methods across several data sets, models and few-shot settings. Our code is publicly available at `https://github.com/oveilleux/Realistic_Transductive_Few_Shot`.

## 1 Introduction

Deep learning models are widely dominating the field. However, their outstanding performances are often built upon training on large-scale labeled data sets, and the models are seriously challenged when dealing with novel classes that were not seen during training. Few-shot learning [1, 2, 3] tackles this challenge, and has recently triggered substantial interest within the community. In standard few-shot settings, a model is initially trained on large-scale data containing labeled examples from a set of *base* classes. Then, supervision for a new set of classes, which are different from those seen in the base training, is restricted to just one or a few labeled samples per class. Model generalization is evaluated over few-shot *tasks*. Each task includes a *query* set containing unlabeled samples for evaluation, and is supervised by a *support* set containing a few labeled samples per new class.

The recent few-shot classification literature is abundant and widely dominated by convoluted meta-learning and episodic-training strategies. To simulate generalization challenges at test times, such strategies build sequences of artificially balanced few-shot tasks (or episodes) during base training, each containing both query and support samples. Widely adopted methods within this paradigm include: Prototypical networks [4], which optimizes the log-posteriors of the query points within each base-training episode; Matching networks [3], which expresses the predictions of query points

---

*Equal contributions, corresponding authors: {olivier.veilleux.2, malik.boudiaf.1}@ens.etsmtl.ca

as linear functions of the support labels, while deploying episodic training and memory architectures; MAML (Model-Agnostic Meta-Learning) [5], which encourages a model to be "easy" to fine-tune; and the meta-learner in [6], which prescribes optimization as a model for few-shot learning. These popular methods have recently triggered a large body of few-shot learning literature, for instance, [7, 8, 9, 10, 11, 12, 13], to list a few.

Recently, a large body of works investigated *transductive* inference for few-shot tasks, e.g., [11, 14, 12, 15, 16, 17, 18, 19, 20, 21, 22, 23], among many others, showing substantial improvements in performances over *inductive* inference[2]. Also, as discussed in [24], most meta-learning approches rely critically on transductive batch normalization (TBN) to achieve competitive performances, for instance, the methods in [5, 25, 26], among others. Adopted initially in the widely used MAML [5], TBN performs normalization using the statistics of the query set of a given few-shot task, and yields significant increases in performances [24]. Therefore, due to the popularity of MAML, several meta-learning techniques have used TBN. The transductive setting is appealing for few-shot learning, and the outstanding performances observed recently resonate well with a well-known fact in classical transductive inference [27, 28, 29]: On small labeled data sets, transductive inference outperforms its inductive counterpart. In few-shot learning, transductive inference has access to exactly the same training and testing data as its inductive counterpart[3]. The difference is that it classifies all the unlabeled query samples of each single few-shot task jointly, rather than one sample at a time.

The current few-shot benchmarks use perfectly class-balanced tasks at inference: For each task used at testing, all the classes have exactly the same number of samples, i.e., the marginal probability of the classes is assumed to be known and fixed to the uniform distribution across all tasks. This may not reflect realistic scenarios, in which testing tasks might come with arbitrary class proportions. For instance, the unlabeled query set of a task could be highly imbalanced. In fact, using perfectly balanced query sets for benchmarking the models assumes exact knowledge of the marginal distributions of the true labels of the testing points, but such labels are unknown. This is, undeniably, an unrealistic assumption and an important limitation of the current few-shot classification benchmarks and datasets. Furthermore, this suggests that the recent progress in performances might be, in part, due to class-balancing priors (or biases) that are encoded in state-of-the-art transductive models. Such priors might be implicit, e.g., through carefully designed episodic-training schemes and specialized architectures, or explicit, e.g., in the design of transductive loss functions and constraints. For instance, the best performing methods in [23, 31] use explicit label-marginal terms or constraints, which strongly enforce perfect class balance within the query set of each task. In practice, those class-balance priors and assumptions may limit the applicability of the existing few-shot benchmarks and methods. In fact, our experiments show that, over few-shot tasks with random class balance, the performances of state-of-the-art methods may decrease by margins. This motivates re-considering the existing benchmarks and re-thinking the relevance of class-balance biases in state-of-the-art methods.

**Contributions**    We introduce and study the effect of arbitrary class distributions within the query sets of few-shot tasks at inference. Specifically, we relax the assumption of perfectly balanced query sets and model the marginal probabilities of the classes as Dirichlet-distributed random variables. We devise a principled procedure for sampling simplex vectors from the Dirichlet distribution, which is widely used in Bayesian statistics for modeling categorical events. This leverages the current few-shot benchmarks by generating testing tasks with arbitrary class distributions, thereby reflecting realistic scenarios. We evaluate experimentally state-of-the-art transductive few-shot methods over 3 widely used datasets, and observe that the performances decrease by important margins, albeit at various degrees, when dealing with arbitrary class distributions. In some cases, the performances drop even below the inductive baselines, which are not affected by class-distribution variations (as they do not use the query-set statistics). Furthermore, we propose a generalization of the transductive mutual-information loss, based on $\alpha$-divergences, which can handle effectively class-distribution variations. Empirically, we show that our transductive $\alpha$-divergence optimization outperforms state-of-the-art few-shot methods across different data sets, models and few-shot settings.

---

[2]The best-performing state-of-the-art few-shot methods in the transductive-inference setting have achieved performances that are up to 10% higher than their inductive counterparts; see [23], for instance.

[3]Each single few-shot task is treated independently of the other tasks in the transductive-inference setting. Hence, the setting does not use additional unlabeled data, unlike semi-supervised few-shot learning [30].

## 2 Standard few-shot settings

**Base training** Assume that we have access to a fully labelled *base* dataset $\mathcal{D}_{base} = \{\boldsymbol{x}_i, \boldsymbol{y}_i\}_{i=1}^{N_{base}}$, where $\boldsymbol{x}_i \in \mathcal{X}_{base}$ are data points in an input space $\mathcal{X}_{base}$, $\boldsymbol{y}_i \in \{0,1\}^{|\mathcal{Y}_{base}|}$ the one-hot labels, and $\mathcal{Y}_{base}$ the set of base classes. Base training learns a feature extractor $f_{\boldsymbol{\phi}} : \mathcal{X} \to \mathcal{Z}$, with $\boldsymbol{\phi}$ its learnable parameters and $\mathcal{Z}$ a (lower-dimensional) feature space. The vast majority of the literature adopts episodic training at this stage, which consists in formatting $\mathcal{D}_{base}$ as a series of tasks (=episodes) in order to mimic the testing stage, and train a meta-learner to produce, through a differentiable process, predictions for the query set. However, it has been repeatedly demonstrated over the last couple years that a standard supervised training followed by standard transfer learning strategies actually outperforms most meta-learning based approaches [32, 33, 34, 20, 23]. Therefore, we adopt a standard cross-entropy training in this work.

**Testing** The model is evaluated on a set of few-shot tasks, each formed with samples from $\mathcal{D}_{test} = \{\boldsymbol{x}_i, \boldsymbol{y}_i\}_{i=1}^{N_{test}}$, where $\boldsymbol{y}_i \in \{0,1\}^{|\mathcal{Y}_{test}|}$ such that $\mathcal{Y}_{base} \cap \mathcal{Y}_{test} = \emptyset$. Each task is composed of a labelled support set $\mathcal{S} = \{\boldsymbol{x}_i, y_i\}_{i \in I_{\mathcal{S}}}$ and an unlabelled query set $\mathcal{Q} = \{\boldsymbol{x}_i\}_{i \in I_{\mathcal{Q}}}$, both containing instances only from $K$ distinct classes randomly sampled from $\mathcal{Y}_{test}$, with $K < |\mathcal{Y}_{test}|$. Leveraging a feature extractor $f_{\boldsymbol{\phi}}$ pre-trained on the base data, the objective is to learn, for each few-shot task, a classifier $f_{\boldsymbol{W}} : \mathcal{Z} \to \Delta_K$, with $\boldsymbol{W}$ the learnable parameters and $\Delta_K = \{\boldsymbol{y} \in [0,1]^K \;/\; \sum_k y_k = 1\}$ the $(K-1)$-simplex. To simplify the equations for the rest of the paper, we use the following notations for the posterior predictions of each $i \in I_{\mathcal{S}} \cup I_{\mathcal{Q}}$ and for the class marginals within $\mathcal{Q}$:

$$p_{ik} = f_{\boldsymbol{W}}(f_{\boldsymbol{\phi}}(\boldsymbol{x}_i))_k = \mathbb{P}(Y = k | X = \boldsymbol{x}_i; \boldsymbol{W}, \boldsymbol{\phi}) \text{ and } \widehat{p}_k = \frac{1}{|I_{\mathcal{Q}}|} \sum_{i \in I_{\mathcal{Q}}} p_{ik} = \mathbb{P}(Y_{\mathcal{Q}} = k; \boldsymbol{W}, \boldsymbol{\phi}),$$

where $X$ and $Y$ are the random variables associated with the raw features and labels, respectively; $X_{\mathcal{Q}}$ and $Y_{\mathcal{Q}}$ means restriction of the random variable to set $\mathcal{Q}$. The end goal is to predict the classes of the unlabeled samples in $\mathcal{Q}$ for each few-shot task, independently of the other tasks. A large body of works followed a transductive-prediction setting, e.g., [11, 14, 12, 15, 16, 17, 18, 19, 20, 21, 22, 23], among many others. Transductive inference performs a joint prediction for all the unlabeled query samples of each single few-shot task, thereby leveraging the query-set statistics. On the current benchmarks, tranductive inference often outperforms substantially its inductive counterpart (i.e., classifying one sample at a time for a given task). Note that our method is agnostic to the specific choice of classifier $f_{\boldsymbol{W}}$, whose parameters are learned at inference. In the experimental evaluations of our method, similarly to [23], we used $p_{ik} \propto \exp(-\frac{\tau}{2} \|\boldsymbol{w}_k - \boldsymbol{z}_i\|^2)$, with $\boldsymbol{W} := (\boldsymbol{w}_1, \ldots, \boldsymbol{w}_K)$, $\boldsymbol{z}_i = \frac{f_{\boldsymbol{\phi}}(\boldsymbol{x}_i)}{\|f_{\boldsymbol{\phi}}(\boldsymbol{x}_i)\|_2}$, $\tau$ is a temperature parameter and base-training parameters $\boldsymbol{\phi}$ are fixed[4].

**Perfectly balanced vs imbalanced tasks** In standard $K$-way few-shot settings, the support and query sets of each task $\mathcal{T}$ are formed using the following procedure: (i) Randomly sample $K$ classes $\mathcal{Y}_{\mathcal{T}} \subset \mathcal{Y}_{test}$; (ii) For each class $k \in \mathcal{Y}_{\mathcal{T}}$, randomly sample $n_k^{\mathcal{S}}$ support examples, such that $n_k^{\mathcal{S}} = |\mathcal{S}|/K$; and (iii) For each class $k \in \mathcal{Y}_{\mathcal{T}}$, randomly sample $n_k^{\mathcal{Q}}$ query examples, such that $n_k^{\mathcal{Q}} = |\mathcal{Q}|/K$. Such a setting is undeniably artificial as we assume $\mathcal{S}$ and $\mathcal{Q}$ have the same perfectly balanced class distribution. Several recent works [35, 36, 37, 38] studied class imbalance exclusively on the support set $\mathcal{S}$. This makes sense as, in realistic scenarios, some classes might have more labelled samples than others. However, even these works rely on the assumption that query set $\mathcal{Q}$ is perfectly balanced. We argue that such an assumption is not realistic, as one typically has even less control over the class distribution of $\mathcal{Q}$ than it has over that of $\mathcal{S}$. For the labeled support $\mathcal{S}$, the class distribution is at least fully known and standard strategies from imbalanced supervised learning could be applied [38]. This does not hold for $\mathcal{Q}$, for which we need to make class predictions at testing time and whose class distribution is unknown. In fact, generating perfectly balanced tasks at test times for benchmarking the models assumes that one has access to the unknown class distributions of the query points, which requires access to their unknown labels. More importantly, artificial balancing of $\mathcal{Q}$ is implicitly or explicitly encoded in several transductive methods, which use the query set statistics to make class predictions, as will be discussed in section 4.

---

[4]$\boldsymbol{\phi}$ is either fixed, e.g., [23], or fine-tuned during inference, e.g., [15]. There is, however, evidence in the literature that freezing $\boldsymbol{\phi}$ yields better performances [23, 32, 34, 33], while reducing the inference time.

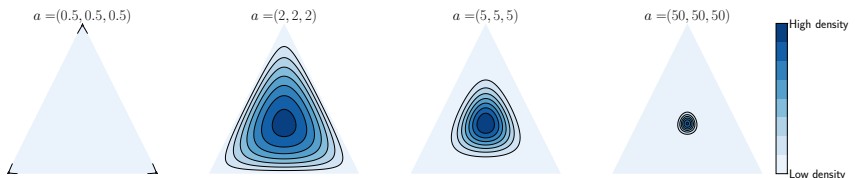

Figure 1: Dirichlet density function for $K = 3$, with different choices of parameter vector $\boldsymbol{a}$.

## 3 Dirichlet-distributed class marginals for few-shot query sets

Standard few-shot settings assume that $p_k$, the proportion of the query samples belonging to a class $k$ within a few-shot task, is deterministic (fixed) and known *priori*: $p_k = n_k^{\mathcal{Q}}/|\mathcal{Q}| = 1/K$, for all $k$ and all few-shot tasks. We propose to relax this unrealistic assumption, and to use the Dirichlet distribution to model the proportions (or marginal probabilities) of the classes in few-shot query sets as random variables. Dirichlet distributions are widely used in Bayesian statistics to model $K$-way categorical events[5]. The domain of the Dirichlet distribution is the set of $K$-dimensional discrete distributions, i.e., the set of vectors in $(K-1)$-simplex $\Delta_K = \{\boldsymbol{p} \in [0,1]^K \mid \sum_k p_k = 1\}$. Let $P_k$ denotes a random variable associated with class probability $p_k$, and $P$ the random simplex vector given by $P = (P_1, \ldots, P_K)$. We assume that $P$ follows a Dirichlet distribution with parameter vector $\boldsymbol{a} = (a_1, \ldots, a_K) \in \mathbb{R}^K$: $P \sim \texttt{Dir}(\boldsymbol{a})$. The Dirichlet distribution has the following density function: $f_{\texttt{Dir}}(\boldsymbol{p}; \boldsymbol{a}) = \frac{1}{B(\boldsymbol{a})} \prod_{k=1}^{K} p_k^{a_k - 1}$ for $\boldsymbol{p} = (p_1, \ldots, p_K) \in \Delta_K$, with $B$ denoting the multivariate Beta function, which could be expressed with the Gamma function[6]: $B(\boldsymbol{a}) = \frac{\prod_{k=1}^{K} \Gamma(a_k)}{\Gamma\left(\sum_{k=1}^{K} \alpha_k\right)}$.

Figure 1 illustrates the Dirichlet density for $K = 3$, with a 2-simplex support represented with an equilateral triangle, whose vertices are probability vectors $(1,0,0)$, $(0,1,0)$ and $(0,0,1)$. We show the density for $\boldsymbol{a} = a\mathbb{1}_K$, with $\mathbb{1}_K$ the $K$-dimensional vector whose all components are equal to 1 and concentration parameter $a$ equal to 0.5, 2, 5 and 50. Note that the limiting case $a \to +\infty$ corresponds to the standard settings with perfectly balanced tasks, where only uniform distribution, i.e., the point in the middle of the simplex, could occur as marginal distribution of the classes.

The following result, well-known in the literature of random variate generation [39], suggests that one could generate samples from the multivariate Dirichlet distribution via simple and standard univariate Gamma generators.

**Theorem 3.1.** *([39, p. 594]) Let $N_1, \ldots, N_K$ be $K$ independent Gamma-distributed random variables with parameters $a_k$: $N_k \sim \texttt{Gamma}(a_k)$, i.e., the probability density of $N_k$ is univariate Gamma[7], with shape parameter $a_k$. Let $P_k = \frac{N_k}{\sum_{k=1}^{K} N_k}$, $k = 1, \ldots, K$. Then, $P = (P_1, \ldots, P_K)$ is Dirichlet distributed: $P \sim \texttt{Dir}(\boldsymbol{a})$, with $\boldsymbol{a} = (a_1, \ldots, a_K)$.*

A proof based on the Jacobian of random-variable transformations $P_k = \frac{N_k}{\sum_{k=1}^{K} N_k}$, $k = 1, \ldots, K$, could be found in [39], p. 594. This result prescribes the following simple procedure for sampling random simplex vectors $(p_1, \ldots, p_K)$ from the multivariate Dirichlet distribution with parameters $\boldsymbol{a} = (a_1, \ldots, a_K)$: First, we draw $K$ independent random samples $(n_1, \ldots, n_K)$ from Gamma distributions, with each $n_k$ drawn from univariate density $f_{\texttt{Gamma}}(n; a_k)$; To do so, one could use standard random generators for the univariate Gamma density; see Chapter 9 in [39]. Then, we set $p_k = \frac{n_k}{\sum_{k=1}^{K} n_k}$. This enables to generate randomly $n_k^{\mathcal{Q}}$, the number of samples of class $k$ within query set $\mathcal{Q}$, as follows: $n_k^{\mathcal{Q}}$ is the closest integer to $p_k |\mathcal{Q}|$ such that $\sum_k n_k^{\mathcal{Q}} = |\mathcal{Q}|$.

---

[5]Note that the Dirichlet distribution is the conjugate prior of the categorical and multinomial distributions.

[6]The Gamma function is given by: $\Gamma(a) = \int_0^\infty t^{a-1} \exp(-t)dt$ for $a > 0$. Note that $\Gamma(a) = (a-1)!$ when $a$ is a strictly positive integer.

[7]Univariate Gamma density is given by: $f_{\texttt{Gamma}}(n; a_k) = \frac{n^{a_k - 1} \exp(-n)}{\Gamma(a_k)}$, $n \in \mathbb{R}$.

# 4 On the class-balance bias of the best-performing few-shot methods

As briefly evoked in section 2, the strict balancing of the classes in both $\mathcal{S}$ and $\mathcal{Q}$ represents a strong inductive bias, which few-shot methods can either meta-learn during training or leverage at inference. In this section, we explicitly show how such a class-balance prior is encoded in the two best-performing transductive methods in the literature [23, 31], one based on mutual-information maximization [23] and the other on optimal transport [31].

**Class-balance bias of optimal transport**    Recently, the transductive method in [31], referred to as PT-MAP, achieved the best performances reported in the literature on several popular benchmarks, to the best of our knowledge. However, the method explicitly embeds a class-balance prior. Formally, the objective is to find, for each few-shot task, an optimal mapping matrix $M \in \mathbb{R}_+^{|\mathcal{Q}| \times K}$, which could be viewed as a joint probability distribution over $X_\mathcal{Q} \times Y_\mathcal{Q}$. At inference, a hard constraint $M \in \{M : M\mathbb{1}_K = r, \mathbb{1}_{|\mathcal{Q}|}M = c\}$ for some $r$ and $c$ is enforced through the use of the Sinkhorn-Knopp algorithm. In other words, the columns and rows of $M$ are constrained to sum to pre-defined vectors $r \in \mathbb{R}^{|\mathcal{Q}|}$ and $c \in \mathbb{R}^K$. Setting $c = \frac{1}{K}\mathbb{1}_K$ as done in [31] ensures that $M$ defines a valid joint distribution, *but also crucially encodes the strong prior that all the classes within the query sets are equally likely*. Such a hard constraint is detrimental to the performance in more realistic scenarios where the class distributions of the query sets could be arbitrary, and not necessarily uniform. Unsurprisingly, PT-MAP undergoes a substantial performance drop in the realistic scenario with Dirichlet-distributed class proportions, with a consistent decrease in accuracy between 18 and 20 % on all benchmarks, in the 5-ways case.

**Class-balance bias of transductive mutual-information maximization**    Let us now have a closer look at the mutual-information maximization in [23]. Following the notations introduced in section 2, the transductive loss minimized in [23] for a given few-shot task reads:

$$\mathcal{L}_{\text{TIM}} = \text{CE} - \mathcal{I}(X_\mathcal{Q}; Y_\mathcal{Q}) = \text{CE} \underbrace{- \frac{1}{|I_\mathcal{Q}|} \sum_{i \in \mathcal{Q}} \sum_{k=1}^{K} p_{ik}\log(p_{ik})}_{\mathcal{H}(Y_\mathcal{Q}|X_\mathcal{Q})} + \lambda \underbrace{\sum_{k=1}^{K} \widehat{p}_k \log \widehat{p}_k}_{-\mathcal{H}(Y_\mathcal{Q})}, \tag{1}$$

where $\mathcal{I}(X_\mathcal{Q}; Y_\mathcal{Q}) = -\mathcal{H}(Y_\mathcal{Q}|X_\mathcal{Q}) + \lambda\mathcal{H}(Y_\mathcal{Q})$ is a weighted mutual information between the query samples and their unknown labels (the mutual information corresponds to $\lambda = 1$), and $\text{CE} := -\frac{1}{|I_\mathcal{S}|} \sum_{i \in \mathcal{S}} \sum_{k=1}^{K} y_{ik}\log(p_{ik})$ is a supervised cross-entropy term defined over the support samples. Let us now focus our attention on the label-marginal entropy term, $\mathcal{H}(Y_\mathcal{Q})$. As mentioned in [23], this term is of significant importance as it prevents trivial, single-class solutions stemming from minimizing only conditional entropy $\mathcal{H}(Y_\mathcal{Q}|X_\mathcal{Q})$. However, we argue that this term also encourages class-balanced solutions. In fact, we can write it as an explicit KL divergence, which penalizes deviation of the label marginals within a query set from the uniform distribution:

$$\mathcal{H}(Y_\mathcal{Q}) = -\sum_{k=1}^{K} \widehat{p}_k \log(\widehat{p}_k) = \log(K) - \mathcal{D}_{\text{KL}}(\widehat{p}\|\mathbf{u}_K). \tag{2}$$

Therefore, minimizing marginal entropy $\mathcal{H}(Y_\mathcal{Q})$ is equivalent to minimizing the KL divergence between the predicted marginal distribution $\widehat{p} = (\widehat{p}_1, \ldots, \widehat{p}_K)$ and uniform distribution $\mathbf{u}_K = \frac{1}{K}\mathbb{1}_K$. This KL penalty could harm the performances whenever the class distribution of the few-shot task is no longer uniform. In line with this analysis, and unsurprisingly, we observe in section 6 that the original model in [23] also undergoes a dramatic performance drop, up to $20\%$. While naively removing this marginal-entropy term leads to even worse performances, we observe that simply down-weighting it, i.e., decreasing $\lambda$ in Eq. (1), can drastically alleviate the problem, in contrast to the case of optimal transport where the class-balance constraint is enforced in a hard manner.

# 5 Generalizing mutual information with $\alpha$-divergences

In this section, we propose a non-trivial, but simple generalization of the mutual-information loss in (1), based on $\alpha$-divergences, which can tolerate more effectively class-distribution variations. We identified in section 4 a class-balance bias encoded in the marginal Shannon entropy term. Ideally,

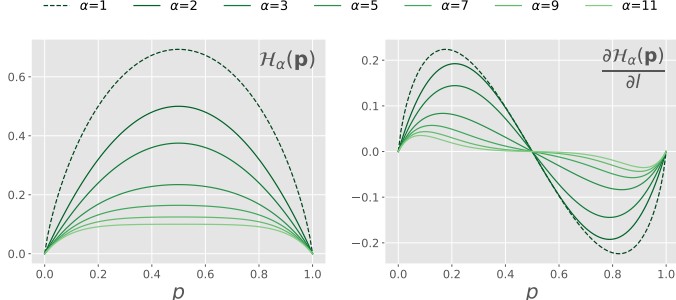

Figure 2: (Left) $\alpha$-entropy as a function of $p = \sigma(l)$. (Right) Gradient of $\alpha$-entropy w.r.t to the logit $l \in \mathbb{R}$ as a function of $p = \sigma(l)$. Best viewed in color.

we would like to extend this Shannon-entropy term in a way that allows for more flexibility: Our purpose is to control how far the predicted label-marginal distribution, $\widehat{p}$, could depart from the uniform distribution without being heavily penalized.

## 5.1 Background

We argue that such a flexibility could be controlled through the use of $\alpha$-divergences [40, 41, 42, 43, 44], which generalize the well-known and widely used KL divergence. $\alpha$-divergences form a whole family of divergences, which encompasses Tsallis and Renyi $\alpha$-divergences, among others. In this work, we focus on Tsallis's [40, 43] formulation of $\alpha$-divergence. Let us first introduce the generalized logarithm [44]: $\log_\alpha(x) = \frac{1}{1-\alpha}\left(x^{1-\alpha} - 1\right)$. Using the latter, Tsallis $\alpha$-divergence naturally extends KL. For two discrete distributions $\boldsymbol{p} = (p_k)_{k=1}^K$ and $\boldsymbol{q} = (q_k)_{k=1}^K$, we have:

$$\mathcal{D}_\alpha(\boldsymbol{p}\|\boldsymbol{q}) = -\sum_{k=1}^K p_k \log_\alpha\left(\frac{q_k}{p_k}\right) = \frac{1}{1-\alpha}\left(1 - \sum_{k=1}^K p_k^\alpha q_k^{1-\alpha}\right). \tag{3}$$

Note that the Shannon entropy in Eq. (2) elegantly generalizes to Tsallis $\alpha$-entropy:

$$\mathcal{H}_\alpha(\boldsymbol{p}) = \log_\alpha(K) - K^{1-\alpha}\,\mathcal{D}_\alpha(\boldsymbol{p}\|\mathbf{u}_K) = \frac{1}{\alpha-1}\left(1 - \sum_k p_k^\alpha\right). \tag{4}$$

The derivation of Eq. (4) is provided in appendix. Also, $\lim_{\alpha\to 1}\log_\alpha(x) = \log(x)$, which implies:

$$\lim_{\alpha\to 1}\mathcal{D}_\alpha(\boldsymbol{p}\|\boldsymbol{q}) = \mathcal{D}_{\mathrm{KL}}(\boldsymbol{p}\|\boldsymbol{q}) \quad \text{and} \quad \lim_{\alpha\to 1}\mathcal{H}_\alpha(\boldsymbol{p}) = \mathcal{H}(\boldsymbol{p}) = -\sum_{k=1}^K \widehat{p}_k \log\left(\widehat{p}_k\right).$$

Note that $\alpha$-divergence $\mathcal{D}_\alpha(\boldsymbol{p}\|\boldsymbol{q})$ inherits the nice properties of the KL divergence, including but not limited to convexity with respect to both $\boldsymbol{p}$ and $\boldsymbol{q}$ and strict positivity $\mathcal{D}_\alpha(\boldsymbol{p}\|\boldsymbol{q}) \geq 0$ with equality if $\boldsymbol{p} = \boldsymbol{q}$. Furthermore, beyond its link to the forward KL divergence $\mathcal{D}_{\mathrm{KL}}(\boldsymbol{p}\|\boldsymbol{q})$, $\alpha$-divergence smoothly connects several well-known divergences, including the reverse KL divergence $\mathcal{D}_{\mathrm{KL}}(\boldsymbol{q}\|\boldsymbol{p})$ ($\alpha \to 0$), the Hellinger ($\alpha = 0.5$) and the Pearson Chi-square ($\alpha = 2$) distances [44].

## 5.2 Analysis of the gradients

As observed from Eq. (4), $\alpha$-entropy is, just like Shannon Entropy, intrinsically biased toward the uniform distribution. Therefore, we still have not properly answered the question: why would $\alpha$-entropy be better suited to imbalanced situations? We argue the that learning dynamics subtly but crucially differ. To illustrate this point, let us consider a simple toy logistic-regression example. Let $l \in \mathbb{R}$ denotes a logit, and $p = \sigma(l)$ the corresponding probability, where $\sigma$ stands for the usual sigmoid function. The resulting probability distribution simply reads $\boldsymbol{p} = \{p, 1-p\}$. In Figure 2, we plot both the $\alpha$-entropy $\mathcal{H}_\alpha$ (left) and its gradients $\partial\mathcal{H}_\alpha/\partial l$ (right) as functions of $p$. The advantage of $\alpha$-divergence now becomes clearer: as $\alpha$ increases, $\mathcal{H}_\alpha(p)$ accepts more and more deviation from the uniform distribution ($p = 0.5$ on Figure 2), while still providing a *barrier* preventing trivial solutions (i.e., $p = 0$ or $p = 1$, which corresponds to all the samples predicted as 0 or 1). Intuitively, such a behavior makes $\alpha$-entropy with $\alpha > 1$ better suited to class imbalance than Shannon entropy.

## 5.3 Proposed formulation

In light of the previous discussions, we advocate a new $\alpha$-mutual information loss, a simple but very effective extension of the Shannon mutual information in Eq. (1):

$$\mathcal{I}_\alpha(X_\mathcal{Q}; Y_\mathcal{Q}) = \mathcal{H}_\alpha(Y_\mathcal{Q}) - \mathcal{H}_\alpha(Y_\mathcal{Q}|X_\mathcal{Q}) = \frac{1}{\alpha - 1} \left( \frac{1}{|I_\mathcal{Q}|} \sum_{i \in I_\mathcal{Q}} \sum_{k=1}^{K} p_{ik}^\alpha - \sum_{k=1}^{K} \widehat{p}_k^\alpha \right) \quad (5)$$

with $\mathcal{H}_\alpha$ the $\alpha$-entropy as defined in Eq. (4). Note that our generalization in Eq. (5) has no link to the $\boldsymbol{\alpha}$-mutual information derived in [45]. Finally, our loss for transductive few-shot inference reads:

$$\mathcal{L}_{\alpha\text{-TIM}} = \text{CE} - \mathcal{I}_\alpha(X_\mathcal{Q}; Y_\mathcal{Q}). \quad (6)$$

## 6 Experiments

In this section, we thoroughly evaluate the most recent few-shot transductive methods using our imbalanced setting. Except for SIB [16] and LR-ICI [17] all the methods have been reproduced in our common framework. All the experiments have been executed on a single GTX 1080 Ti GPU.

**Datasets** We use three standard benchmarks for few-shot classification: *mini*-Imagenet [46], *tiered*-Imagenet [30] and *Caltech-UCSD Birds 200* (*CUB*) [47]. The *mini*-Imagenet benchmark is a subset of the ILSVRC-12 dataset [46], composed of 60,000 color images of size 84 x 84 pixels [3]. It includes 100 classes, each having 600 images. In all experiments, we used the standard split of 64 base-training, 16 validation and 20 test classes [6, 33]. The *tiered*-Imagenet benchmark is a larger subset of ILSVRC-12, with 608 classes and 779,165 color images of size $84 \times 84$ pixels. We used a standard split of 351 base-training, 97 validation and 160 test classes. The *Caltech-UCSD Birds 200* (*CUB*) benchmark also contains images of size $84 \times 84$ pixels, with 200 classes. For CUB, we used a standard split of 100 base-training, 50 validation and 50 test classes, as in [32]. It is important to note that for all the splits and data-sets, the base-training, validation and test classes are all different.

**Task sampling** We evaluate all the methods in the 1-shot 5-way, 5-shot 5-way, 10-shot 5-way and 20-shot 5-way scenarios, with the classes of the query sets randomly distributed following Dirichlet's distribution, as described in section 3. Note that the total amount of query samples $|\mathcal{Q}|$ remains fixed to 75. All the methods are evaluated by the average accuracy over 10,000 tasks, following [33]. We used different Dirichlet's concentration parameter $\boldsymbol{a}$ for validation and testing. The validation-task generation is based on a random sampling within the simplex (i.e Dirichlet with $\boldsymbol{a} = \mathbb{1}_K$). Testing-task generation uses $\boldsymbol{a} = 2 \cdot \mathbb{1}_K$ to reflect the fact that extremely imbalanced tasks (i.e., only one class is present in the task) are unlikely to happen in practical scenarios; see Figure 1 for visualization.

**Hyper-parameters** Unless identified as directly linked to a class-balance bias, all the hyper-parameters are kept similar to the ones prescribed in the original papers of the reproduced methods. For instance, the marginal entropy in TIM [23] was identified in section 4 as a penalty that encourages class balance. Therefore, the weight controlling the relative importance of this term is tuned. For all methods, hyper-parameter tuning is performed on the validation set of each dataset, using the validation sampling described in the previous paragraph.

**Base-training procedure** All non-episodic methods use the same feature extractors, which are trained using the same procedure as in [23, 20], via a standard cross-entropy minimization on the base classes with label smoothing. The feature extractors are trained for 90 epochs, using a learning rate initialized to 0.1 and divided by 10 at epochs 45 and 66. We use a batch size of 256 for ResNet-18 and of 128 for WRN28-10. During training, color jitter, random croping and random horizontal flipping augmentations are applied. For episodic/meta-learning methods, given that each requires a specific training, we use the pre-trained models provided with the GitHub repository of each method.

### 6.1 Main results

The main results are reported in Table 1. As baselines, we also report the performances of state-of-the-art inductive methods that do not use the statistics of the query set at adaptation and are, therefore,

Table 1: Comparisons of state-of-the-art methods in our realistic setting on *mini*-ImageNet, *tiered*-ImageNet and CUB. Query sets are sampled following a Dirichlet distribution with $a = 2 \cdot \mathbb{1}_K$. Accuracy is averaged over 10,000 tasks. A red arrow ($\downarrow$) indicates a performance drop between the artificially-balanced setting and our testing procedure, and a blue arrow ($\uparrow$) an improvement. Arrows are not displayed for the inductive methods as, for these, there is no significant change in performance between both settings (expected). '–' signifies the result was computationally intractable to obtain.

| | Method | Network | *mini*-ImageNet | | | |
| | | | 1-shot | 5-shot | 10-shot | 20-shot |
|---|---|---|---|---|---|---|
| Induct. | Protonet (NeurIPS'17 [4]) | RN-18 | 53.4 | 74.2 | 79.2 | 82.4 |
| | Baseline (ICLR'19 [32]) | | 56.0 | 78.9 | 83.2 | 85.9 |
| | Baseline++ (ICLR'19 [32]) | | 60.4 | 79.7 | 83.8 | 86.3 |
| | Simpleshot (arXiv [33]) | | 63.0 | 80.1 | 84.0 | 86.1 |
| Transduct. | MAML (ICML'17 [5]) | RN-18 | 47.6 ($\downarrow$3.8) | 64.5 ($\downarrow$5.0) | 66.2 ($\downarrow$5.7) | 67.2 ($\downarrow$3.6) |
| | Versa (ICLR'19 [25]) | | 47.8 ($\downarrow$2.2) | 61.9 ($\downarrow$3.7) | 65.6 ($\downarrow$3.6) | 67.3 ($\downarrow$4.0) |
| | Entropy-min (ICLR'20 [15]) | | 58.5 ($\downarrow$5.1) | 74.8 ($\downarrow$7.3) | 77.2 ($\downarrow$8.0) | 79.3 ($\downarrow$7.9) |
| | LR+ICI (CVPR'2020 [17]) | | 58.7 ($\downarrow$8.1) | 73.5 ($\downarrow$5.7) | 78.4 ($\downarrow$2.7) | 82.1 ($\downarrow$1.7) |
| | PT-MAP (arXiv [31]) | | 60.1 ($\downarrow$16.8) | 67.1 ($\downarrow$18.2) | 68.8 ($\downarrow$18.0) | 70.4 ($\downarrow$17.4) |
| | LaplacianShot (ICML'20 [20]) | | 65.4 ($\downarrow$4.7) | 81.6 ($\downarrow$0.5) | 84.1 ($\downarrow$0.2) | 86.0 ($\uparrow$0.5) |
| | BD-CSPN (ECCV'20 [21]) | | 67.0 ($\downarrow$2.4) | 80.2($\downarrow$1.8) | 82.9 ($\downarrow$1.4) | 84.6 ($\downarrow$1.1) |
| | TIM (NeurIPS'20 [23]) | | 67.3 ($\downarrow$4.5) | 79.8 ($\downarrow$4.1) | 82.3 ($\downarrow$3.8) | 84.2 ($\downarrow$3.7) |
| | $\alpha$-TIM (ours) | | **67.4** | **82.5** | **85.9** | **87.9** |
| Induct. | Baseline (ICLR'19 [32]) | WRN | 62.2 | 81.9 | 85.5 | 87.9 |
| | Baseline++ (ICLR'19 [32]) | | 64.5 | 82.1 | 85.7 | 87.9 |
| | Simpleshot (arXiv [33]) | | 66.2 | 82.4 | 85.6 | 87.4 |
| Transduct. | Entropy-min (ICLR'20 [15]) | WRN | 60.4 ($\downarrow$5.7) | 76.2 ($\downarrow$8.0) | – | – |
| | PT-MAP (arXiv [31]) | | 60.6 ($\downarrow$18.3) | 66.8 ($\downarrow$19.8) | 68.5 ($\downarrow$19.3) | 69.9 ($\downarrow$19.0) |
| | SIB (ICLR'20 [16]) | | 64.7 ($\downarrow$5.3) | 72.5 ($\downarrow$6.7) | 73.6 ($\downarrow$8.4) | 74.2 ($\downarrow$8.7) |
| | LaplacianShot (ICML'20 [20]) | | 68.1 ($\downarrow$4.8) | 83.2 ($\downarrow$0.6) | 85.9 ($\uparrow$0.4) | 87.2 ($\uparrow$0.6) |
| | TIM (NeurIPS'20 [23]) | | 69.8 ($\downarrow$4.8) | 81.6 ($\downarrow$4.3) | 84.2 ($\downarrow$3.9) | 85.9 ($\downarrow$3.7) |
| | BD-CSPN (ECCV'20 [21]) | | **70.4** ($\downarrow$2.1) | 82.3($\downarrow$1.4) | 84.5 ($\downarrow$1.4) | 85.7 ($\downarrow$1.1) |
| | $\alpha$-TIM (ours) | | 69.8 | **84.8** | **87.9** | **89.7** |

| | Method | Network | *tiered*-ImageNet | | | |
| | | | 1-shot | 5-shot | 10-shot | 20-shot |
|---|---|---|---|---|---|---|
| Induct. | Baseline (ICLR'19 [32]) | RN-18 | 63.5 | 83.8 | 87.3 | 89.0 |
| | Baseline++ (ICLR'19 [32]) | | 68.0 | 84.2 | 87.4 | 89.2 |
| | Simpleshot (arXiv [33]) | | 69.6 | 84.7 | 87.5 | 89.1 |
| Transduct. | Entropy-min (ICLR'20 [15]) | RN-18 | 61.2 ($\downarrow$5.8) | 75.5 ($\downarrow$7.6) | 78.0 ($\downarrow$7.9) | 79.8 ($\downarrow$7.9) |
| | PT-MAP (arXiv [31]) | | 64.1 ($\downarrow$18.8) | 70.0 ($\downarrow$18.8) | 71.9 ($\downarrow$17.8) | 73.4 ($\downarrow$17.1) |
| | LaplacianShot (ICML'20 [20]) | | 72.3 ($\downarrow$4.8) | 85.7 ($\downarrow$0.5) | 87.9 ($\downarrow$0.1) | 89.0 ($\uparrow$0.3) |
| | BD-CSPN (ECCV'20 [21]) | | 74.1 ($\downarrow$2.2) | 84.8 ($\downarrow$1.4) | 86.7 ($\downarrow$1.1) | 87.9 ($\downarrow$0.8) |
| | TIM (NeurIPS'20 [23]) | | 74.1 ($\downarrow$4.5) | 84.1 ($\downarrow$3.6) | 86.0 ($\downarrow$3.3) | 87.4 ($\downarrow$3.1) |
| | LR+ICI (CVPR'20 [17]) | | **74.6** ($\downarrow$6.2) | 85.1 ($\downarrow$2.8) | 88.0 ($\downarrow$2.1) | 90.2 ($\downarrow$1.2) |
| | $\alpha$-TIM (ours) | | 74.4 | **86.6** | **89.3** | **90.9** |
| Induct. | Baseline (ICLR'19 [32]) | WRN | 64.6 | 84.9 | 88.2 | 89.9 |
| | Baseline++ (ICLR'19 [32]) | | 68.7 | 85.4 | 88.4 | 90.1 |
| | Simpleshot (arXiv [33]) | | 70.7 | 85.9 | 88.7 | 90.1 |
| Transduct. | Entropy-min (ICLR'20 [15]) | WRN | 62.9 ($\downarrow$6.0) | 77.3 ($\downarrow$7.5) | – | – |
| | PT-MAP (arXiv [31]) | | 65.1 ($\downarrow$19.5) | 71.0 ($\downarrow$19.0) | 72.5 ($\downarrow$18.3) | 74.0 ($\downarrow$17.7) |
| | LaplacianShot (ICML'20 [20]) | | 73.5 ($\downarrow$5.3) | 86.8 ($\downarrow$0.5) | 88.6 ($\downarrow$0.4) | 89.6 ($\downarrow$0.2) |
| | BD-CSPN (ECCV'20 [21]) | | 75.4 ($\downarrow$2.3) | 85.9 ($\downarrow$1.5) | 87.8 ($\downarrow$1.0) | 89.1 ($\downarrow$0.6) |
| | TIM (NeurIPS'20 [23]) | | 75.8 ($\downarrow$4.5) | 85.4 ($\downarrow$3.5) | 87.3 ($\downarrow$3.2) | 88.7 ($\downarrow$2.9) |
| | $\alpha$-TIM (ours) | | **76.0** | **87.8** | **90.4** | **91.9** |

Table 2: Comparaisons of state-of-the-art methods in our realistic setting on CUB. Query sets are sampled following a Dirichlet distribution with $\boldsymbol{a} = 2 \cdot \mathbb{1}_K$. Accuracy is averaged over 10,000 tasks. A red arrow ($\downarrow$) indicates a performance drop between the artificially-balanced setting and our testing procedure, and a blue arrow ($\uparrow$) an improvement. Arrows are not displayed for the inductive methods as, for these, there is no significant change in performance between both settings (expected). '–' signifies the result was computationally intractable to obtain.

| | | | CUB | | | |
| | | | 1-shot | 5-shot | 10-shot | 20-shot |
|---|---|---|---|---|---|---|
| Induct. | Baseline (ICLR'19 [32]) | | 64.6 | 86.9 | 90.6 | 92.7 |
| | Baseline++ (ICLR'19 [32]) | RN-18 | 69.4 | 87.5 | 91.0 | 93.2 |
| | Simpleshot (ARXIV [33]) | | 70.6 | 87.5 | 90.6 | 92.2 |
| Transduct. | PT-MAP (ARXIV [31]) | | 65.1 ($\downarrow$ 20.4) | 71.3 ($\downarrow$ 20.0) | 73.0 ($\downarrow$ 19.2) | 72.2 ($\downarrow$ 18.9) |
| | Entropy-min (ICLR'20 [15]) | | 67.5 ($\downarrow$ 5.3) | 82.9 ($\downarrow$ 6.0) | 85.5 ($\downarrow$ 5.6) | 86.8 ($\downarrow$ 5.7) |
| | LaplacianShot (ICML'20 [20]) | RN-18 | 73.7 ($\downarrow$ 5.2) | 87.7 ($\downarrow$ 1.1) | 89.8 ($\downarrow$ 0.7) | 90.6 ($\downarrow$ 0.5) |
| | BD-CSPN (ECCV'20 [21]) | | 74.5 ($\downarrow$ 3.4) | 87.1 ($\downarrow$ 1.8) | 89.3 ($\downarrow$ 1.3) | 90.3 ($\downarrow$ 1.1) |
| | TIM (NEURIPS'20 [23]) | | 74.8 ($\downarrow$ 5.5) | 86.9 ($\downarrow$ 3.6) | 89.5 ($\downarrow$ 2.9) | 91.7 ($\downarrow$ 2.8) |
| | $\alpha$-TIM (ours) | | **75.7** | **89.8** | **92.3** | **94.6** |

unaffected by class imbalance. In the 1-shot scenario, all the transductive methods, without exception, undergo a significant drop in performances as compared to the balanced setting. Even though the best-performing transductive methods still outperforms the inductive ones, we observe that more than half of the transductive methods evaluated perform overall worse than inductive baselines in our realistic setting. Such a surprising finding highlights that the standard benchmarks, initially developed for the inductive setting, are not well suited to evaluate transductive methods. In particular, when evaluated with our protocol, the current state-of-the-art holder PT-MAP averages more than 18% performance drop across datasets and backbones, Entropy-Min around 7%, and TIM around 4%. Our proposed $\alpha$-TIM method outperforms transductive methods across almost all task formats, datasets and backbones, and is the only method that consistently inductive baselines in fair setting. While stronger inductive baselines have been proposed in the literature [48], we show in the supplementary material that $\alpha$-TIM keeps a consistent relative improvement when evaluated under the same setting.

## 6.2 Ablation studies

**In-depth comparison of TIM and $\alpha$-TIM**    While not included in the main Table 1, keeping the same hyper-parameters for TIM as prescribed in the original paper [23] would result in a drastic drop of about 18% across the benchmarks. As briefly mentioned in section 4 and implemented for tuning [23] in Table 1, adjusting marginal-entropy weight $\lambda$ in Eq. (1) strongly helps in imbalanced scenarios, reducing the drop from 18% to only 4%. However, we argue that such a strategy is sub-optimal in comparison to using $\alpha$-divergences, where the only hyper-parameter controlling the flexibility of the marginal-distribution term becomes $\alpha$. First, as seen from Table 1, our $\alpha$-TIM achieves consistently better performances with the same budget of hyper-parameter optimization as the standard TIM. In fact, in higher-shots scenarios (5 or higher), the performances of $\alpha$-TIM are substantially better that the standard mutual information (*i.e.* TIM). Even more crucially, we show in Figure 3 that $\alpha$-TIM does not fail drastically when $\alpha$ is chosen sub-optimally, as opposed to the case of weighting parameter $\lambda$ for the TIM formulation. We argue that such a robustness makes of $\alpha$-divergences a particularly interesting choice for more practical applications, where such a tuning might be intractable. Our results points to the high potential of $\alpha$-divergences as loss functions for leveraging unlabelled data, beyond the few-shot scenario, e.g., in semi-supervised or unsupervised domain adaptation problems.

**Varying imbalance severity**    While our main experiments used a fixed value $\boldsymbol{a} = 2 \cdot \mathbb{1}_K$, we wonder whether our conclusions generalize to different levels of imbalance. Controlling for Dirichlet's parameter $\boldsymbol{a}$ naturally allows us to vary the imbalance severity. In Figure 4, we display the results obtained by varying $\boldsymbol{a}$, while keeping the same hyper-parameters obtained through our validation protocol. Generally, most methods follow the expected trend: as $\boldsymbol{a}$ decreases and tasks become more severely imbalanced, performances decrease, with sharpest losses for TIM [23] and PT-MAP [31]. In fact, past a certain imbalance severity, the inductive baseline in [33] becomes more competitive than

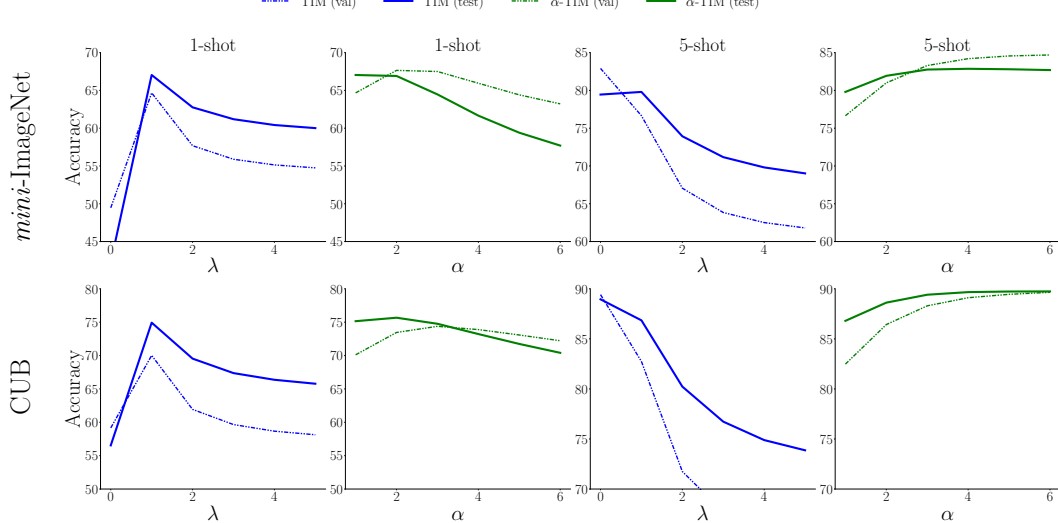

Figure 3: Validation and Test accuracy versus $\lambda$ for TIM [23] and $\alpha$ for our proposed $\alpha$-TIM, using our protocol. Results are obtained with a RN-18. Best viewed in color.

most transductive methods. Interestingly, both LaplacianShot [20] and our proposed $\alpha$-TIM are able to cope with extreme imbalance, while still conserving good performances on balanced tasks.

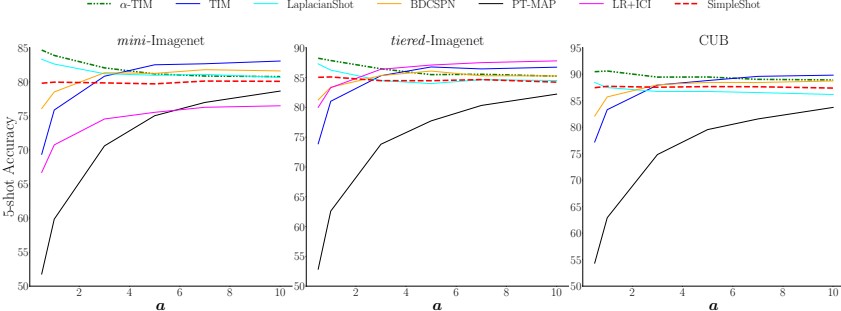

Figure 4: 5-shot test accuracy of transductive methods versus imbalance level (lower $\boldsymbol{a}$ corresponds to more severe imbalance, as depicted in Figure 1).

**On the use of transductive BN**   In the case of imbalanced query sets, we note that transductive batch normalization (e.g as done in the popular MAML [49]) hurts the performances. This aligns with recent observations from the concurrent work in [50], where a shift in the marginal label distribution is clearly identified as a failure case of statistic alignment via batch normalization.

## Conclusion

We make the unfortunate observation that recent transductive few-shot methods claiming large gains over inductive ones may perform worse when evaluated with our realistic setting. The artificial balance of the query sets in the vanilla setting makes it easy for transductive methods to implicitly encode this strong prior at meta-training stage, or even explicitly at inference. When rendering such a property obsolete at test-time, the current top-performing method becomes noncompetitive, and all the transductive methods undergo performance drops. Future works could study the mixed effect of imbalance on both support and query sets. We hope that our observations encourage the community to rethink the current transductive literature, and build upon our work to provide fairer grounds of comparison between inductive and transductive methods.

## Acknowledgments

This project was supported by the Natural Sciences and Engineering Research Council of Canada (Discovery Grant RGPIN 2019-05954). This project has received funding from the European Union's Horizon 2020 research and innovation programme under the Marie Skłodowska-Curie grant agreement №792464.

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
