# Realistic Evaluation of Transductive Few-Shot Learning - Supplementary Material

**Olivier Veilleux** *
ÉTS Montreal

**Malik Boudiaf** *
ÉTS Montreal

**Pablo Piantanida**
L2S, CentraleSupélec CNRS
Université Paris-Saclay

**Ismail Ben Ayed**
ÉTS Montreal

## A    On the performance of $\alpha$-TIM on the standard balanced setting

In the main tables of the paper, we did not include the performances of $\alpha$-TIM in the standard balanced setting. Here, we emphasize that $\alpha$-TIM is a generalization of TIM [1] as when $\alpha \to 1$ (i.e., the $\alpha$-entropies tend to the Shannon entropies), $\alpha$-TIM tends to TIM. Therefore, in the standard setting, where optimal hyper-parameter $\alpha$ is obtained over validation tasks that are balanced (as in the standard validation tasks of the original TIM and the other existing methods), the performance of $\alpha$-TIM is the same as TIM. When $\alpha$ is tuned on balanced validation tasks, we obtain an optimal value of $\alpha$ very close to 1, and our $\alpha$-mutual information approaches the standard mutual information. When the validation tasks are uniformly random, as in our new setting and in the validation plots we provided in the main figure, one can see that the performance of $\alpha$-TIM remains competitive when we tend to balanced testing tasks (i.e., when $a$ is increasing), but is significantly better than TIM when we tend to uniformly-random testing tasks ($a = 1$). These results illustrate the flexibility of $\alpha$-divergences, and are in line with the technical analysis provided in the main paper.

## B    Comparison with DeepEMD

The recent method [2] achieves impressive results in the inductive setting. As conveyed in the main paper, inductive methods tend to be unaffected by class imbalance on the query set, which legitimately questions whether strong inductive methods should be preferred over transductive ones, including our proposed $\alpha$-TIM. In the case of DeepEMD, we expand below on the levers used to obtain such results, and argue those are orthogonal to the loss function, and therefore to our proposed $\alpha$-TIM method. More specifically:

1. DeepEMD uses richer feature representations: While all the methods we reproduce use the standard global average pooling to obtain a single feature vector per image, DeepEMD-FCN leverages dense feature maps (i.e without the pooling layer). This results in a richer, much higher-dimensional embeddings. For instance, the standard RN-18 yields a 512-D vector per image, while the FCN RN-12 used by DeepEMD yields a 5x5x640-D feature map (i.e 31x larger). As for DeepEMD-Grid and DeepEMD-Sampling, they build feature maps by concatenating feature extracted from N different patches taken from the original image (which requires as many forward passes through the backbone). Also, note that prototypes optimized for during inference have the same dimension as the feature maps. Therefore, taking richer and larger feature representations also means increasing the number of trainable parameters at inference by the same ratio.

2. DeepEMD uses a more sophisticated notion of distance (namely the Earth Moving Distance), introducing an EMD layer, different from the standard classification layer. While all methods we reproduced are based on simple and easy-to-compute distances between each feature and the prototypes (e.g Euclidian, dot-product, cosine distance), the flow-based distance used by

---

*Equal contributions, corresponding authors: {olivier.veilleux.2, malik.boudiaf.1}@ens.etsmtl.ca

35th Conference on Neural Information Processing Systems (NeurIPS 2021).

Table 1: Comparison with DeepEMD [2]. Input: **W**=Whole images are used as input ; **P** = Multiples patches of the whole image are used as input. Embeddings: **G**=Global averaged features are used (i.e 1 feature vector per image) ; **L** = Local features are used (i.e 1 feature map per image ).

| Method | Distance | RN-18 (W/G) | WRN (W/G) | FCN RN-12 (W/L) | Grid RN-12 (P/L) | Sampling RN-12 (P/L) |
|--------|----------|-------------|-----------|-----------------|------------------|----------------------|
| SimpleShot [3] | Euclidian | 63.0 | 66.2 | — | — | — |
| $\alpha$-TIM | Euclidian | 67.4 | 69.8 | — | — | — |
| DeepEMD [2] | EMD | — | — | 65.9 | 67.8 | 68.8 |
| $\alpha$-TIM | EMD | — | — | 68.9 | 72.0 | 72.6 |

DeepEMD captures more complex patterns than the usual Euclidian distance, but is also much more demanding computationally (as it requires solving an LP program).

Now, we want to emphasize that the model differences mentioned above can be straightforwardly applied to our $\alpha$-TIM (and likely the other methods) in order to boost the results at the cost of a significant increase of compute requirement. To demonstrate this point, we implemented our $\alpha$-TIM in with the three ResNet-12 based architectures proposed in DeepEMD (cf Table 1) using our imbalanced tasks, and consistently observed +3 to +4 without changing any optimization hyper-parameter from their setting, and using the pre-trained models the authors have provided. This figure matches the improvement observed w.r.t to SimpleShot with the standard models (RN-18 and WRN).

## C  Relation between $\alpha$-entropy and $\alpha$-divergence

We provide the derivation of Eq. (4) in the main paper, which links $\alpha$-entropy $\mathcal{H}_\alpha(\mathbf{p})$ to the $\alpha$-divergence:

$$
\begin{aligned}
\log_\alpha(K) - K^{1-\alpha}\mathcal{D}_\alpha(\mathbf{p}\|\mathbf{u}_K) &= \frac{1}{1-\alpha}\left(K^{1-\alpha} - 1\right) - \frac{K^{1-\alpha}}{\alpha-1}\left(\sum_{k=1}^{K} p_k^\alpha \left(\frac{1}{K}\right)^{1-\alpha} - 1\right) \\
&= \frac{1}{1-\alpha}K^{1-\alpha} - \frac{1}{1-\alpha} - \frac{1}{\alpha-1}\sum_{k=1}^{K} p_k^\alpha + \frac{K^{1-\alpha}}{\alpha-1} \\
&= \frac{1}{\alpha-1}\left(1 - \sum_{k=1}^{K} p_k^\alpha\right)
\end{aligned}
\tag{1}
$$

## D  Comparison with other types of imbalance

The study in [4] examined the effect of class imbalance on the support set after defining several processes to generate class-imbalanced support sets. In particular, the authors proposed *linear* and *step* imbalance. In a 5-way setting, a typical *linearly* imbalanced few-shot support would look like $\{1, 3, 5, 7, 9\}$ (keeping the total number of support samples equivalent to standard 5-ways 5-shot tasks), while a *step* imbalance task could be $\{1, 9, 9, 9\}$. To provide intuition as to how these two types of imbalance related to our proposed Dirichlet-based sampling scheme, we super-impose Dirichlet's density on all valid *linear* and *step* imbalanced distributions for 3-ways tasks in Figure 1. Combined, *linear* and *step* imbalanced valid distributions allow to cover a fair part of the simplex, but Dirichlet sampling allows to sample more diverse and arbitrary class ratios.

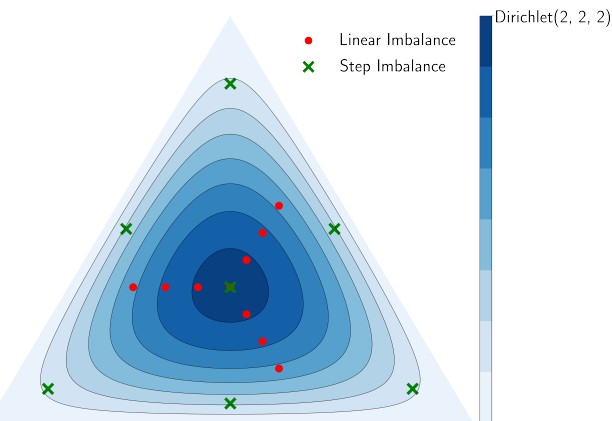

Figure 1: Comparison of Dirichlet sampling with linear and step imbalance.

# E   Influence of each term in TIM and $\alpha$-TIM

We report a comprehensive ablation study, evaluating the benefits of using the $\alpha$-entropy instead of the Shannon entropy (both conditional and marginal terms), as well as the effect of the marginal-entropy terms in the loss functions of TIM and $\alpha$-TIM. The results are reported in Table 2. $\alpha$-TIM yields better performances in all settings.

**On the $\alpha$-conditional entropy:** The results of $\alpha$-TIM obtained by optimizing the conditional entropy alone (without the marginal term) are 4.5 to 7.2% higher in 1-shot, 0.8 to 3.5% higher in 5-shot and 0.1 to 1.3% higher in 10-shot scenarios, in comparison to its Shannon-entropy counterpart in TIM. Note that, for the conditional-probability term, the $\alpha$-entropy formulation has a stronger effect in lower-shot scenarios (1-shot and 5-shot). We hypothesize that this is due to the shapes of the $\alpha$-entropy functions and their gradient dynamics (see Fig. 2 in the main paper), which, during training, assigns more weight to confident predictions near the vertices of the simplex ($p = 1$ or $p = 0$) and less weight to uncertain predictions at the middle of the simplex ($p = 0.5$). This discourages propagation of errors during training (i.e., learning from uncertain predictions), which are more likely to happen in lower-shot regimes.

**Flexibility of the $\alpha$-marginal entropy:** An important observation is that the marginal-entropy term does even hurt the performances of TIM in the higher shot scenarios (10-shot), even though the results correspond to the best $\lambda$ over the validation set. We hypothesize that this is due to the strong class-balance bias in the Shannon marginal entropy. Again, due to the shapes of the $\alpha$-entropy functions and their gradient dynamics, $\alpha$-TIM tolerates better class imbalance. In the 10-shot scenarios, the performances of TIM decrease by 1.8 to 3.2% when including the marginal entropy, whereas the performance of $\alpha$-TIM remains approximately the same (with or without the marginal-entropy term). These performances demonstrate the flexibility of $\alpha$-TIM.

Table 2: An ablation study evaluating the benefits of using the $\alpha$-entropy instead of the Shannon entropy (both conditional and marginal terms), as well as the effect of the marginal-entropy terms in the loss functions of TIM and $\alpha$-TIM.

| Loss | Dataset | Network | Method | 1-shot | 5-shot | 10-shot |
|---|---|---|---|---|---|---|
| $CE + \mathcal{H}(Y_{\mathcal{Q}}|X_{\mathcal{Q}})$ | *mini*-Imagenet | RN-18 | TIM | 42.2 | 79.5 | 85.5 |
| | | | $\alpha$-TIM | **48.4** | **82.4** | **86.0** |
| | | WRN | TIM | 52.8 | 82.7 | 87.5 |
| | | | $\alpha$-TIM | **57.3** | **84.6** | **88.0** |
| | *tiered*-Imagenet | RN-18 | TIM | 52.4 | 83.7 | 88.4 |
| | | | $\alpha$-TIM | **59.0** | **86.3** | **89.2** |
| | | WRN | TIM | 49.6 | 84.1 | 89.1 |
| | | | $\alpha$-TIM | **56.8** | **87.6** | **90.4** |
| | CUB | RN-18 | TIM | 56.4 | 89.0 | 92.2 |
| | | | $\alpha$-TIM | **63.2** | **89.8** | **92.3** |
| $CE + \mathcal{H}(Y_{\mathcal{Q}}|X_{\mathcal{Q}}) - \mathcal{H}(Y_{\mathcal{Q}})$ | *mini*-Imagenet | RN-18 | TIM | 67.3 | 79.8 | 82.3 |
| | | | $\alpha$-TIM | **67.4** | **82.5** | **85.9** |
| | | WRN | TIM | 69.8 | 82.3 | 84.5 |
| | | | $\alpha$-TIM | 69.8 | **84.8** | **87.9** |
| | *tiered*-Imagenet | RN-18 | TIM | 74.1 | 84.1 | 86.0 |
| | | | $\alpha$-TIM | **74.4** | **86.6** | **89.3** |
| | | WRN | TIM | 75.8 | 85.4 | 87.3 |
| | | | $\alpha$-TIM | **76.0** | **87.8** | **90.4** |
| | CUB | RN-18 | TIM | 74.8 | 86.9 | 89.5 |
| | | | $\alpha$-TIM | **75.7** | **89.8** | **92.3** |

# F  Hyper-parameters validation

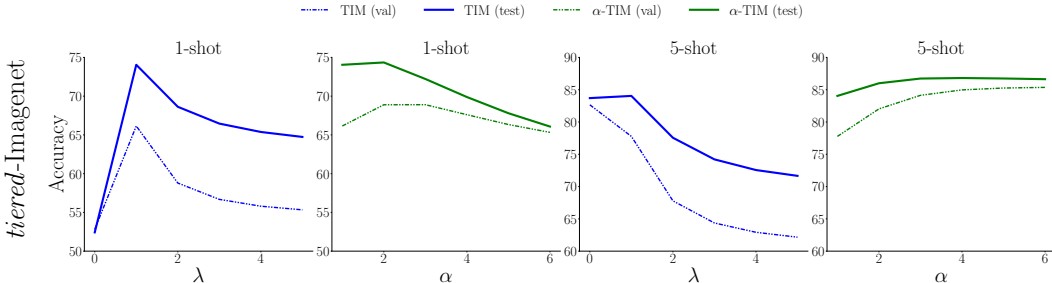

Figure 2: Validation and Test accuracy versus $\lambda$ for TIM [1] and versus $\alpha$ for $\alpha$-TIM, using our task-generation protocol. Results are obtained with a RN-18. Best viewed in color.

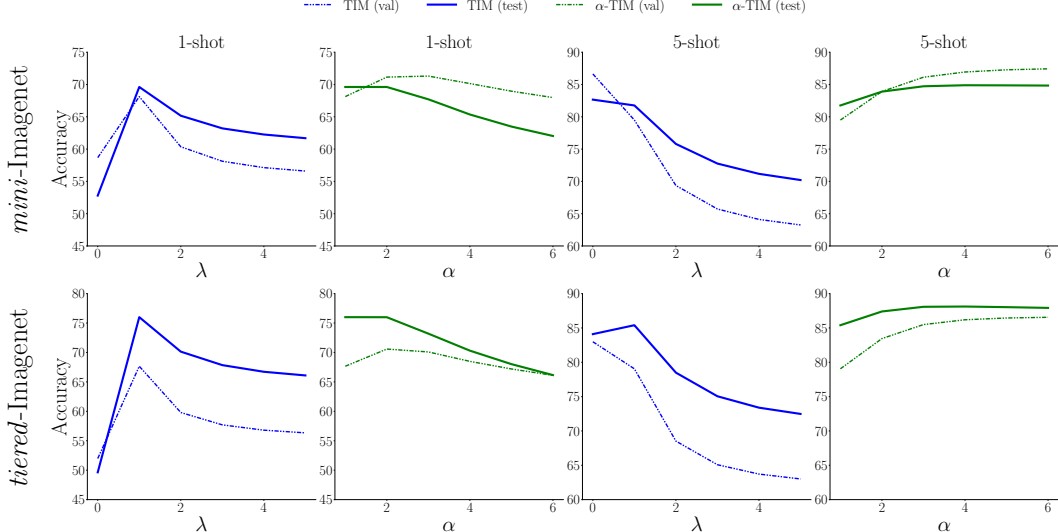

Figure 3: Validation and Test accuracy versus $\lambda$ for TIM [1] and versus $\alpha$ for $\alpha$-TIM, using our task-generation protocol. Results are obtained with a WRN. Best viewed in color.

# G  Code – Implementation of our framework

As mentioned in our main experimental section, all the methods have been reproduced in our common framework, except for SIB[2] [5] and LR-ICI[3] [6], for which we used the official public implementations of the works.

---

[2]SIB public implementation: `https://github.com/hushell/sib_meta_learn`
[3]LR-ICI public implementation: `https://github.com/Yikai-Wang/ICI-FSL`

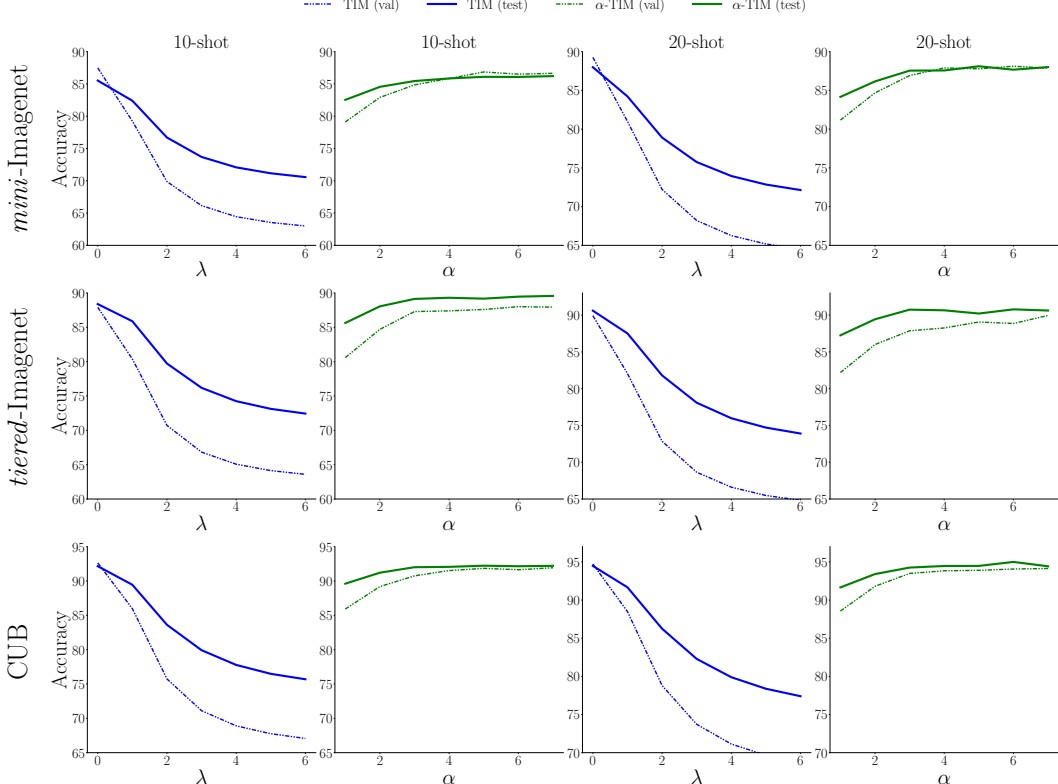

Figure 4: Validation and Test accuracy versus $\lambda$ for TIM [1] and versus $\alpha$ for $\alpha$-TIM on 10-shot and 20-shot tasks, using our task-generation protocol. Results are obtained with a RN-18. Best viewed in color.

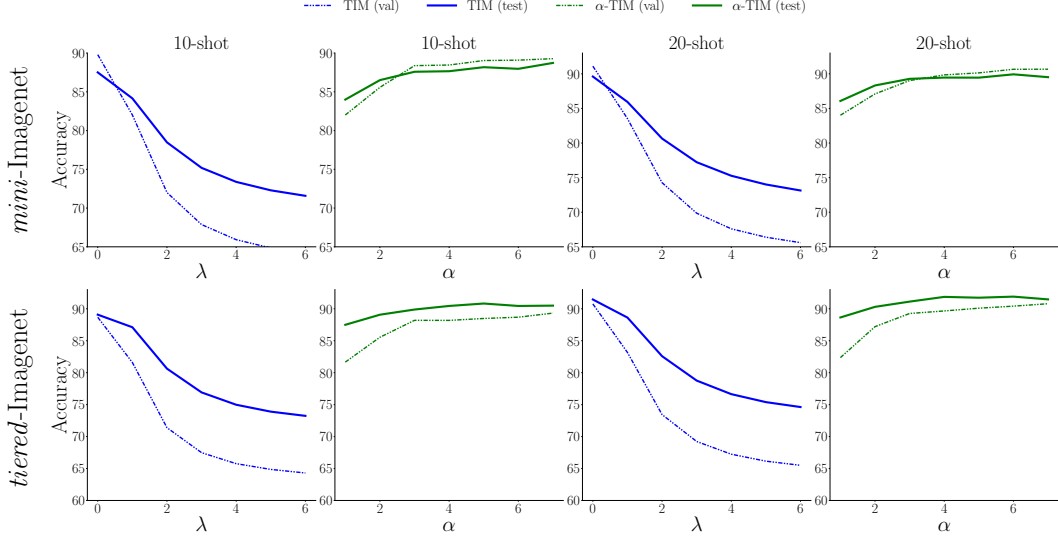

Figure 5: Validation and Test accuracy versus $\lambda$ for TIM [1] and versus $\alpha$ for $\alpha$-TIM on 10-shot and 20-shot tasks, using our task-generation protocol. Results are obtained with a WRN. Best viewed in color.