# OpenReview forum: "Realistic evaluation of transductive few-shot learning"
_NeurIPS.cc/2021/Conference — NeurIPS 2021 Poster_

### Official Review · Reviewer_W251 · 2021-06-26

**Rating:** 7
**Confidence:** 3

**Summary:**

The paper studies transductive few-shot learning in the more realistic setting with arbitrary class distributions within the query sets. In the proposed setting, authors use Dirichlet distribution to model the marginal probabilities of the classes in the query sets as random variables and generate random samples within the simplex. Focusing on two recent transductive methods PT-MAP and TIM, they first show that class-balance prior is encoded in these two methods and then extend TIM by generalizing mutual-information loss to \alpha divergences to more effectively deal with arbitrary class distributions. The experiments validate proposed \alpha-divergence approach on three standard benchmark datasets (mini-ImageNet, tiered-ImageNet and CUB) and show that the method outperforms inductive and transductive few-shot learning methods in the proposed setting.

**Limitations And Societal Impact:**

Limitations of the work are not mentioned. For example, is it better to use existing methods with balanced class distributions (see cons)?

**Main Review:**

Pros:
- The paper is well-motivated, clearly written and deals with a more realistic evaluation of the transductive few-shot learning methods.
- Proposed alpha-divergence approach is a simple and effective extension of TIM for dealing with the arbitrary class distributions of the query
- Experimental results show improvements over existing methods

Cons:
- The paper focuses on only two transductive methods PT-MAP and TIM to show that the class-balance prior is encoded in these methods. What about other transductive methods? For example, from the experiments, BD-CSPN does not seem to be so sensitive (especially under assumption that some performance drop is expected). Why is the paper focused only on transfer learning and not on meta-learning approaches?  The current study is not comprehensive.
- Experimental results are not convincing. First, there is a lack of stronger baselines in Table 1. Authors claim to outperform state-of-the-art inductive methods but do not compare to recent approaches such as MetaOptNet [1], CTM [2] and DeepEMD [3]. The difference between inductive and transductive methods in this setting seems small and with stronger baselines claims might change.
- Why is Dirichlet parameter a set to 2? How does the performance change under different imbalance distributions? Where is the point where existing methods start performing better? Extremely, how does the method compare under a standard balanced scenario? More experiments are needed to answer these questions.
- In Table 1, how is the \alpha parameter tuned? Which values of \alpha are reported in the table? Similarly for TIM, how is \lambda tuned and which values are reported?

[1] Meta-Learning with Differentiable Convex Optimization. CVPR ‘19

[2] Finding Task-Relevant Features for Few-Shot Learning by Category Traversal. CVPR ‘19

[3] DeepEMD: Differentiable Earth Mover’s Distance for Few-Shot Learning. CVPR ‘20


**Time Spent Reviewing:**

5

---

> ### Author Response · Authors · 2021-08-10
> **Hyper-parameter tuning**
>
> 7- *In Table 1, how is the $\alpha$ parameter tuned? Which values of $\alpha$ are reported in the table? Similarly for TIM, how is $\lambda$ tuned and which values are reported?*
>
> As mentioned in lines 261-264, we tune every hyper-parameter (including $\alpha$ and $\lambda$) using validation tasks (with different classes) and Dirichlet parameter $a=1$, which corresponds to uniformly-random tasks. The range of values used to tune $\alpha$ on the validation tasks is from 1 to 10, with a step of 1, while the range to validate $\lambda$ goes from 0 to 2, with a step of 0.25. For each method, the best value obtained on validation tasks is selected. The idea was to keep the hyper-parameter budget similar, as mentioned in line 298. Also, as discussed in lines 291-295,  using the same hyper-parameters for TIM as prescribed in the original paper [23] would result in a drastic drop of about 18% across the benchmarks. Adjusting $\lambda$ with our validation strongly helped TIM in imbalanced scenarios, reducing the drop from 18% to only 4%.

---

> ### Author Response · Authors · 2021-08-10
> **Choice of methods analyzed and reproduced**
>
> 4- *Why is the paper focused only on transfer learning and not on meta-learning approaches? The current study is not comprehensive.*
>
> In the popular transductive setting, which is the main focus of this study, we reported in Table 1 the 6 best-performing and most recent methods in the literature (all 6 from 2020 and 5 published in the top conferences), independently of  whether the methods are based on transfer learning or meta-learning. In fact, in Table 1, LR+ICI (CVPR’2020) and SIB (ICLR 2020) are meta-learning methods, whereas the other 4 are based on transfer learning. Also, we actually evaluated more meta-learning methods that use transductive batch normalization, including the very popular MAML (Finn et al., ICML 2017) and more recent transductive meta-learning methods that built on it like VERSA (Gordon et al., ICLR 2019); the results can be found in Table B. We did not include the results of these methods in the main Table as they are not among the top-6 methods (their performances are significantly lower than the state-of-the-art). However, the conclusion is similar: these meta-learning methods, similarly to the more recent/competitive meta-learning methods we reported in Table 1 (SIB and LR+ICI), are significantly affected by removing the artificial class balance. We will add the results of these methods in the supplemental material (as the main Table is already very big).
>
>
> 5- *The difference between inductive and transductive methods in this setting seems small and with stronger baselines claims might change.*
>
> In fact, we report SimpleShot (also from end of 2019), which has a performance comparable to the three recent methods mentioned by the reviewer (MetaOptNet, CTM and DeepEMD). All inductive methods are still several points below the best-performing transductive methods, even in our new setting. For instance, in (miniImageNet, 1-shot, RN), MetaOptNet reports $62\\%$, CTM $64\\%$ and DeepEMD $65\\%$, whereas Simpleshot in our Table 1 achieves $63\\%$ with RN-18 and $66\\%$ with WRN. The transductive methods achieve up to $70\\%$. In any case, we believe our message would not be altered by having stronger inductive baselines at all. If anything, it would be strengthened, as one of our goals is to show that a more realistic evaluation of transductive methods actually questions their large-margin superiority over inductive methods, which is often claimed in the  few-shot literature.
>
> 6- *The paper focuses on only two transductive methods  to show that the class-balance prior is encoded in these methods. What about other transductive methods?*
>
> We mentioned that meta-learning and episodic training strategies build sequences of artificially balanced
> few-shot tasks (or episodes) during base training (lines 31-33). We also mentioned in several parts of the paper that carefully designed episodic schemes, which uses
> strictly balanced support and query sets during meta-training, might encode implicitly class-balance biases (lines 59-63 and the beginning of section 4). In our experiments, as discussed in our answer above, we evaluated several recent and competitive meta-learning methods and observed significant drops in performances in the new setting. Therefore, as mentioned by the other reviewers, we believe our study is comprehensive in the context of transductive few-shot classification. We concede that the technical analysis of PT-MAP and TIM in Section 4 is longer that the discussions related to meta-learning methods. However, we believe there are strong motivations for that. First, both were, at the time of submission, the two best performing transductive methods, which inevitably makes them relevant candidates. Second, each method utilizes popular regularization techniques in semi-supervised / unsupervised literature, namely mutual-information and optimal transport regularization, and we believe insights derived from analyzing these two methods will actually benefit researchers from a broader community than few-shot only.
>
> As for BD-CSPN, the performance drops by noticeable margins too, although not at the same degree of severity as PT-MAP or TIM, as illustrated in the new
> Fig. A we provide, showing the performances of the top-5 transductive methods versus Dirichlet parameter $\boldsymbol{a}$.

---

> > ### Comment · Reviewer_W251 · 2021-08-16
> > **Re: Follow-up**
> >
> > Thank you for your response. I have follow-up questions.
> >
> > 4. I am aware that you are comparing performance with meta-learning approaches in your experimental results. My question is about the Section 4 where you show that class-balance prior is encoded in the best-performing transductive methods. Can this be shown for other (meta-learning) methods?
> >
> > 5. Simpleshot achieves 63% with ResNet-18, while DeepEMD achieves 69% with ResNet12 on the mini-Imagenet (so the difference would be even higher with ResNet18). On tiered ImageNet, DeepEMD achieves 74% with ResNet12 while SimpleShot acheives 69% with ResNet18.  These differences are larger than the improvements of your method compared to SimpleShot.

---

> > > ### Author Response · Authors · 2021-08-21
> > > **Additional results**
> > >
> > > ## Table C
> > >
> > > All results are produced for 1-shot Dirichlet imbalanced episodes on mini-ImageNet  with $\boldsymbol{a}=2$. Legend:
> > >
> > > Input : **W**=Whole images are used as input ; **P** = Multiples patches of the whole image are used as input
> > >
> > > Embeddings: **G** =Global averaged features are used (i.e one 1 feature vector per image) ; **L** = Local features are used (i.e 1 feature map per image )
> > >
> > > | Method | Distance | RN-18  (W/G)| WRN-2810 (W/G)| FCN RN-12 (W/L)| Grid RN-12 (P/L)| Sampling RN-12 (P/L)|
> > > | ---             | ---      | ---                 | ---                     |---                         |---                   |---                |
> > > | SimpleShot | Euclidian            | 63.0                | 66.2                    |---                         |---                   |---                |
> > > | $\alpha$-TIM | Euclidian         | 67.4                | 69.8                    |---                         |---                   |---                |
> > > | DeepEMD | EMD |                      ---                  |  ---                      | 65.9                       | 67.8                 | 68.8              |
> > > | $\alpha$-TIM | EMD            |  ---                  |  ---                      |  68.9                         | 72.0                 | 72.6              |
> > >
> > >
> > > ## Table D
> > >
> > >  Both methods are compared on the same Tesla-P100 GPU, averaged over 1000 5-way 5-shot episodes.
> > >
> > > | Method        | Network | Average Runtime per episode (s)  |
> > > | ---           |    ---  |   ---              |
> > > | $\alpha$-TIM | WRN-2810 | $0.7 \times 10^0$    |
> > > | DeepEMD | FCN RN-12  | $1.2 \times 10^{+1}$    |

---

> > > ### Author Response · Authors · 2021-08-21
> > > **Answer to 4.**
> > >
> > > We cannot show it explicitly as it is not encoded explicitly in the transductive loss functions (unlike the best-performing transfer-learning methods). However, we hypothesize that standard episodic-training strategies, which create balanced support and query sets during meta-training (for convenience and to simplify meta-training), encode implicitly class-balance biases. While this intuition is strongly confirmed with our experiments with several meta-learning methods, we do not claim to provide formal proofs, as this would probably require a full-fledged theoretical paper for each meta-learning method. In fact, along this interesting theoretical line, but for a different bias created by the number of shots during meta-training, the recent work of [Cao et al., A Theoretical Analysis of the Number of Shots in Few-Shot Learning, ICLR 2020] focused a whole paper on a theoretical analysis of the impact of the shot number on the Prototypical-Networks method. Such a theoretical analysis explained experimental observations showing that the choice of the shot number in meta-training create a bias during meta-testing (the shot number in meta-training should match the one used in meta-testing to obtain the best performance).

---

> > > ### Author Response · Authors · 2021-08-21
> > > **Answer to 5.**
> > >
> > > ### Preamble
> > >
> > >  Thanks for the follow-up questions. Below, (i) We clarify in details the discrepancies in the different figures given for DeepEMD due to changes in the inputs (patches), denser (higher-dimensional) feature representations and more sophisticated distance (with changes in the architecture and number of trainable parameters), (ii) We provide new results (cf Table C below) showing that such model and input changes can be straightforwardly applied to $\alpha$-TIM (and likely most methods) to boost the results at the cost of an increase of compute requirement: Under the same model/inputs, $\alpha$-TIM outperforms deepEMD by a consistent 3 to 4 percent, which matches the improvement observed over SimpleShot when using the standard models and inputs (this confirms that, under the same setting, transductive methods outperforms their inductive counterparts); (iii) clarify why we mentioned that SimpleShot is on par with the state-of-the-art.
> > >
> > > We hope that our answer below clarifies that the DeepEMD model is not in direct competition with $\alpha$-TIM (it could be used jointly with $\alpha$-TIM, replacing standard ResNet-18 or WRN models), and that the differences in performances due to model and input changes are orthogonal to the most significant message of our work (i.e., the gains obtained by transductive methods are over-estimated in the literature). So, stronger inductive baseline/model architectures could only strengthen our main message, and we will add the performances of DeepEMD architectures to the paper (thanks for pointing to the DeepEMD model!).
> > >
> > > ### (i) Clarifications on the different DeepEMD performance figures and different inputs/models:
> > >
> > > First, we clarify that the 65.90\% mini-Imagenet performance for DeepEMD mentioned in our initial answer is based on the standard whole-image input, and is the official result published in the CVPR version [2]. As for the 69\% mentioned by reviewer W251, it refers to new results provided in Table 3 of the pre-print journal extension of DeepEMD [1] when using  multiple image patches as inputs (instead of the whole image). Similarly, $\alpha$-TIM performance increases by about the same 3\% with those patch-based inputs; please see the difference between $\alpha$-TIM (FCN RN12) and $\alpha$-TIM (Sampling RN-12) in the Table C below. The inputs for those new Sampling and Grid versions are several image patches, each encoded individually by a CNN, yielding a different (higher-dimensional) feature embedding concatenating the feature embeddings of all the patches.
> > >
> > > Now, we clarify how all the three setting considered in DeepEMD differ from the standard models (ResNet-18 and WRN) used in few-shot methods:
> > >
> > > 1) DeepEMD uses richer feature representations: While all the methods we reproduce use the standard global average pooling to obtain a single feature vector per image, DeepEMD-FCN leverages dense feature maps (the authors removed the pooling layers).
> > > This results in a richer, much higher-dimensional embeddings. For instance, the standard RN-18 yields a 512-D vector per image, while the FCN RN-12 used by DeepEMD yields a 5x5x640-D feature map (i.e **31x bigger**). As for DeepEMD-Grid and DeepEMD-Sampling, they build feature maps by concatenating feature extracted from N different patches taken from the original image (which, by the way, requires as many forward passes through the backbone). Also, note that prototypes optimized for during inference have the same dimension as the feature maps. Therefore, taking richer and larger feature representations also means increasing the number of trainable parameters at inference by the same ratio.
> > >
> > > 2) DeepEMD uses a more sophisticated notion of distance, introducing an EMD layer that is different from the standard classification layers: While all methods we reproduced are based on simple/easy-to-compute distance between feature and prototypes (e.g Euclidian, dot-product, cosine distance), the flow-based distance used by DeepEMD proposed captures more complex patterns than usual Euclidian distance, but is also much more demanding computationally.
> > >
> > > ### (ii) Such changes in inputs and models (higher-dimensional embeddings, different architecture with EMD layer and no average pooling) also benefits the other methods
> > >
> > > Now, we want to emphasize that the model differences mentioned above can be straightforwardly applied to our $\alpha$-TIM (and likely the other methods) in order to boost the results at the cost of a significant increase of compute requirement. To demonstrate this point, we implemented our $\alpha$-TIM in with the three ResNet-12 based architectures proposed in DeepEMD (cf Table C below) using our imbalanced tasks, and consistently observed +3 to +4 $\%$ without changing any optimization hyper-parameter from their setting, and using the pre-trained models provided in their official GitHub. This figure matches the improvement observed w.r.t to SimpleShot with the standard models (RN-18 and WRN). It also confirms the expected fact that transductive methods outperforms their inductive counterpart when evaluated under the same models/inputs. Also, to make our point on the increased compute requirement, we provide a comparison of the run-time of $\alpha$-TIM + WRN and DeepEMD with ResNet-12 (cf Table G below), with a striking difference of one order of magnitude in favor or $\alpha$-TIM + WRN.
> > >
> > > ### (iii) DeepEMD beats SimpleShot significantly (5 and 6\%) with ResNet12 on mini-Imagenet/tieredImageNet, and the performance of DeepEMD should increase with ResNet-18
> > >
> > > We kindly disagree, and provide below additional numbers/clarifications. As discussed above,
> > > the performances of Grid and Sampling deepEMD correspond to different patch-based inputs and changes in the model architecture and, therefore, are not directly comparable to SimpleShot using basic ResNet-18 with whole-image inputs. In fact, as shown in the Table below, using DeepEMD-FCN would increase the performances of $\alpha$-TIM if used instead of basic ResNet-18, almost by the same margin WRN increases the performances w.r.t RN-18; please refer to (WRN + $\alpha$-TIM) vs. (FCN RN12 + $\alpha$-TIM) in the Table below. Note that DeepEMD scales poorly to larger feature maps yielded by larger backbones (e.g., WRN), which may explain why DeepEMD only presents results with ResNet-12 (5x5x640 maps) rather than the more standard ResNet-18. In the Table below, we provide the inference times (per episode) for DeepEMD-FCN and WRN, which shows that DeepEMD-FCN is significantly slower (illustrating the fact that the model is more complex than basic ResNet-12).
> > >
> > > In principle, one should compare methods under the same models. However, when comparing performance numbers with different architectures (basic ResNet-18 in SimpleShot and $\alpha$-TIM versus fully convolutional DeepEMD-FCN in DeepEMD), and if we allow DeepEMD-FCN to be more complex than basic ResNet architectures, we should also look at the performance of the best model (higher-capacity WRN) for both SimpleShot and $\alpha$-TIM (otherwise it is not fair). Please note that, for the same whole-image inputs, SimpleShot+WRN is on par with DeepEMD-FCN (while being much more efficient computationally at inference), and this is why we mentioned that SimpleShot is on par with inductive state-of-the-art performances. Also, $\alpha$-TIM + WRN is on par with $\alpha$-TIM + DeepEMD-FCN, which points to the fact that DeepEMD-FCN is a more complex model than the basic ResNet-12.

---

> ### Author Response · Authors · 2021-08-10
> **On the imbalance setting**
>
> 1- *Why is Dirichlet parameter a set to 2?*
>
> We refer to answer 2- in https://openreview.net/forum?id=6ns5QTPQ_d&noteId=heqVu9cRDrV made to reviewer maGM.
>
> 2- *How does the performance change under different imbalance distributions?*
>
> We refer to answer 1- in https://openreview.net/forum?id=6ns5QTPQ_d&noteId=heqVu9cRDrV made to reviewer maGM.
>
> 3- *Extremely, how does the method compare under a standard balanced scenario?*
>
> We refer to answer 6- in https://openreview.net/forum?id=6ns5QTPQ_d&noteId=RC5bqsNPmxa made to reviewer maGM.

---

> ### Author Response · Authors · 2021-08-10
> **Preamble**
>
> We thank the reviewer for his comments. In addition to answering the comments, we provide the requested additional results in this rebuttal. Please note that figures indexed with a letter (e.g Fig. A) can be found in the following anonymous link: https://github.com/anonymous6496/rebuttal, while figures indexed with a number are references to figures of the main paper.

---

### Official Review · Reviewer_maGM · 2021-07-14

**Rating:** 7
**Confidence:** 5

**Summary:**

This paper evaluates several transductive few-shot learning methods [15,28,20,21,23,17] on class-imbalanced FSL tasks. The class distribution in the query set follows a Dirichlet distribution. The support set is balanced. This work proposes a novel method based on $\alpha$-divergences addressing the imbalance problem. The novel method $\alpha$-TIM achieves competitive performance compared to several recent baselines.

**Limitations And Societal Impact:**

Some limitations were listed, but the paper would benefit from a separate limitation section.

**Main Review:**

\+ means strength, \- means weakness

**Originality**

\+ *Novel setting.* This paper presents a novel, more realistic setting for the evaluation of transductive learning methods. Unlike previous work on few-shot imbalance [32, 33, 34, 35], this paper focuses on the imbalance present in the query set.

\+ *Novel algorithm.* This paper presents an augmentation of the TIM algorithm [23] that combines with $\alpha$-divergences [37, 38, 39, 40, 41]. This algorithm achieves state-of-the-art performances compared to several recent state-of-the-art transductive few-shot methods [15,28,20,21,23,17]. Unlike this previous work, the authors apply a unique mechanism to deal with class imbalance.

**Quality**

\+ *The paper compares a range of recent methods, datasets and backbones.* Several recent transductive [15,28,20,21,23,17] and non-transductive [4, 29, 30] algorithms are evaluated in this novel setting. The datasets included Mini-ImageNet, tiered-ImageNet, and CUB. And backbones included WRN and ResNet-18.

\- *Evaluation on only one distribution is limiting.* The authors use one alpha value for the Dirichlet distribution (lines 257-264) to model query set imbalance during evaluation. However, it is not clear why this distribution was picked over others. The class imbalance community commonly a range of distributions, for example, step imbalance and linear imbalance (Buda et al., 2018), long-tail imbalance (Wang et al., 2018), and in few-shot previous work sampled uniformly at random [32,33,34,35], or a combination of these distributions [35]. It might be appropriate to evaluate several distributions or vary the $a$ in the Dirichlet distribution.

\- *The optimal $\alpha$ in $\alpha$-TIM might depend on the Dirichlet distribution and the shot setting.* Figure 3 shows that in the 1-shot setting, the optimal $\alpha$ value for the test dataset is around the value of the $\alpha=2$ similar to that concentration parameter used in the Dirichlet distribution ($a=2$). It could be interesting to show the correlation between $\alpha$ and $a$ in a graph. The authors could also examine a wider $\alpha$ range as the values for the 5-shot settings sometimes do not appear optimal. What are the optimal $\alpha$'s for higher shot settings (e.g. 10-shot, 20-shot)? Is there a correlation between the $\alpha$ and shots?

\- *Missing values in Table 1.* What is $alpha$-TIM performance on the balanced setting? For completeness and clarity, for methods [4, 29, 30], it might be worth adding a symbol (similar to the red/blue arrow) to indicate no change in performance. Also, there is no performance for [15] in 10-shot and 20-shot settings with mini-ImageNet and tiered ImageNet. Was that intentional?

**Clarity**

\+/\- *The paper is generally well organised but some clarifications and adjustments would help.* Some adjustments would make this work easier to follow. For instance, the bullet points for the contributions section. The paper should emphasise earlier that imbalance is studied only at the query set level, and support sets are assumed to be balanced. Some sentences could be improved, split or condensed, e.g. lines 81-83.

\- *Clarification for eq. (4).* Why do the authors multiply the $\mathcal{D}_\alpha (p||u_K)$ by the term $K^{1-\alpha}$ in eq. (4)? The term does not appear in the RHS of eq. (2). In addition, it might be worth adding a derivation in the appendix.

\- *Clarification about control and imbalance of the query set (Q).* The authors claim (in lines 121-122) that recent studies [32, 33, 34, 35] "rely on" keeping the query set balanced. However, I can't entirely agree, and I would argue that for *non-transductive* few-shot classification methods such as those studied in [32, 33, 34, 35], the distribution of Q does not matter during meta-testing. During the meta-training (assuming full access to the base dataset), Q can be controlled arbitrarily. Keeping a balanced Q is usually done for convenience but this does not restrict "non-transductive" methods. However, I do agree that for *transductive* methods assuming that the query set is balanced and accessible all at once is unrealistic, and will directly affect performance (as evidenced in this paper) so studying this area seems important.

\- *No $\lambda$ in mutual information tradeoff?*. Eq. (5) does not contain the $\lambda$ in eq. (1). What is the reasoning behind this? What is the performance when $\lambda$ is included and tuned with $\alpha$?

**Significance**

\+ *The paper presents a more realistic evaluation of transductive methods.* This analysis shows that several relevant and recent transductive few-shot methods underperform under query set imbalance. These results would be both interesting and valuable to the community.

\+ *The paper presents a novel and interesting approach to the problem.* The paper presents an interesting take on the class imbalance problem through the $\alpha$-divergences. Making the method $\alpha$-TIM significantly different from TIM and having the most significant improvements in the higher shot settings.

\- *However, the paper feels like work in progress and could be improved significantly* (as outlined here). I am inclined towards a borderline reject at the moment, but I look forward to hearing back from the authors and seeing how this work progresses in the future.

**Some spelling mistakes**

\- line 133, "priori" --> "a prior"

\- line 283, "outperforms" --> "outperform"

\- line 304, "results points" --> "result points" / "results point"

**References**

Buda et al., 2018, "A systematic study of the class imbalance problem in convolutional neural networks." Neural Networks 106 (2018): 249-259.

Wang et al., 2017, "Learning to Model the Tail." NeurIPS 2017


_____
### POST-REBUTTAL
_____
The authors have answered all of my questions above expectations through additional experiments and clear explanations. I will now recommend this paper for the conference with a score of 7 and increased confidence (conditioned on the inclusion of the additional experiments in the main paper / supplementary materials).


**Time Spent Reviewing:**

7

---

> ### Author Response · Authors · 2021-08-10
> **Miscellaneous clarifications**
>
> 8- *Claim on recent studies [32, 33, 34, 35] and inductive methods*
>
>  We fully agree with the reviewer. In fact, we point this out in l.278-280, stating that inductive methods do not use the statistics of the query set at adaptation and are, therefore, unaffected by class imbalance. We also concede that the wording of l.121 was misleading, and we will modify it. "Even these works" should in fact read "Even [32]", as [32] evaluates both the popular MAML and a transductive variant  (transductive through the use of transductive batch normalization), which could be affected by changes in the query-set balance (as shown in Table B provided in this rebuttal).
>
> 9- *Clarification for eq. (4)*
>
>  The right-hand side of Eq. (4) is the definition of Tsallis $\alpha$-entropy. From the term in the middle in Eq. (4), one can recover the definition using the following steps:
>
> \begin{align}
> \log_{\alpha}(K) - K^{1-\alpha} \mathcal{D}_{\alpha} ( \mathbf{p} || \mathbf{u}_K ) &= \frac{1}{1-\alpha} \left( K^{1-\alpha}-1 \right) - \frac{K^{1-\alpha}}{\alpha-1} \left( \sum_k p_k^{\alpha} ( \frac{1}{K})^{1-\alpha} - 1 \right) \\\\
> &= \frac{1}{1 - \alpha} K^{1-\alpha} - \frac{1}{1 - \alpha}  - \frac{1}{\alpha-1} \sum_k p_k^{\alpha} + \frac{1}{\alpha-1}K^{1-\alpha} \\\\
> &= \frac{1 }{\alpha - 1} \left(1 - \sum_k p_k^{\alpha} \right) \\\\
> & = \mathcal{H}_\alpha ( \mathbf{p} )
> \end{align}
>
> The term $K^{1-\alpha}$ naturally disappears when $\alpha \rightarrow 1$, which is why it does not appear in the original Shannon Entropy in Eq (2). We will include this derivation in the final version of the paper.
>
> 10- *There is no performance for [15] in 10-shot and 20-shot settings with mini-ImageNet and tiered ImageNet*
>
>  In fact, we provided the performance of [15] in the 10-shot and 20-shot settings for the RN backbone (5th line in Table 1). For the WRN backbone, as mentioned at the end of the caption of Table 1, results of [15] were intractable to obtain within standard GPU hardware compute. Indeed, method [15] requires fine-tuning the whole network, which we found manageable for ResNet-18 memory-wise, but intractable for WRN and larger tasks. However, we do not expect the trend to be any different from the one observed on ResNet-18.

---

> ### Author Response · Authors · 2021-08-10
> **Performance of $\alpha$-TIM**
>
> 4- *It could be interesting to show the correlation between $\alpha$ and $\boldsymbol{a}$ in a graph.*
>
>
> We plot in Fig. D the optimal value of $\alpha$ (on validation) for different values of $\boldsymbol{a}$. We observe an expected behavior: As $\mathbf{a}$ increases (i.e the query sets in testing become more and more balanced), the optimal value $\alpha^*$ decreases and tend to 1 (which approaches the standard mutual information), with a trend that closely matches $\alpha^*=\frac{\text{const.}}{a} + 1$.
>
>
>  5- *Correlation between $\alpha$ and the number of shots? What are the optimal’s for higher shot settings (e.g. 10-shot, 20-shot)?*
>
>  As illustrated in Fig. E, the behaviour for 10-shot and 20-shot is quite similar to 5-shot, with the performances reaching a plateau when $\alpha$ goes past a certain value.
>
>  6- *What is $\alpha$-TIM performance on the balanced setting?*
>
>  $\alpha$-TIM is a generalization of TIM, as when $\alpha \rightarrow 1$ (i.e., the $\alpha$-entropies tend to the Shannon entropies), $\alpha$-TIM tends to TIM. Therefore, in the standard setting, where optimal hyper-parameter $\alpha$ is obtained over validation tasks that are balanced (as in the standard validation tasks of the original TIM and the other existing methods), the performance of $\alpha$-TIM is the same as TIM. When $\alpha$ is tuned on balanced validation tasks, we obtain an optimal value of $\alpha$ very close to 1, and our $\alpha$-mutual information approaches the standard mutual information. We will add comments on this in the paper.  When the validation tasks are uniformly random, as in our new setting and in the performance plots we provide here, one can see that the performance of $\alpha$-TIM remains competitive when we tend to balanced testing tasks (i.e., when $a$ is increasing), but is significantly better than TIM when we tend to uniformly-random testing tasks ($a=1$). These results illustrate the flexibility of $\alpha$-divergences, and are in line with the technical analysis we provided in section 5.2.
>
>  7- *Reasoning behind removing $\lambda$ for Eq. 5? What is the performance when it is included and tuned ?*
>
>
> As mentioned by Reviewer KiFQ, we wanted to maintain equal hyper-parameter tuning budget with TIM for fairer comparison. $\alpha$ and $\lambda$ hyper-parameters can be understood as two different ways of dealing with class imbalance, with $\alpha$ being more felxible as it modifies the very shape of Shannon entropy's gradient curve, while $\lambda$ only scales it. While including both hyper-parameters would be interesting and would surely lead to better results, we wanted to keep the proposed formulation as simple and practically usable as possible, and only use one lever to deal with imbalance.

---

> ### Author Response · Authors · 2021-08-10
> **Additional results**
>
> ## Table A: 5-shot with RN-18
>
> #### *Mini*-Imagenet
> | Method        | $\boldsymbol{a}= 0.5$ |  $\boldsymbol{a}= 1$ |   $\boldsymbol{a}= 3$ |   $\boldsymbol{a}= 5$ |   $\boldsymbol{a}= 7$ | $\boldsymbol{a}= 10$  |
> | ---           |  ---  | --- | --- | --- | --- | --- |
> | PT-MAP        |  51.8 | 59.8| 70.6| 75.0| 77.0| 78.6|
> | BDCSPN        | 76.1  | 78.5| 81.3| 81.2| 81.8| 81.6|
> | LaplacianShot | 83.3  | 82.6| 81.2| 81.0| 81.1| 80.7|
> | LR+ICI        | 66.7  | 70.7| 74.5| 75.5| 76.3| 76.5|
> | TIM           | 69.4  | 75.9| 80.8| 82.5| 82.7| 83.1|
> | $\alpha$-TIM  | 84.7  | 83.9| 82.1| 81.2| 80.9| 80.8|
>
> #### *Tiered*-Imagenet
> | Method        | $\boldsymbol{a}= 0.5$ |  $\boldsymbol{a}= 1$ |   $\boldsymbol{a}= 3$ |   $\boldsymbol{a}= 5$ |   $\boldsymbol{a}= 7$ | $\boldsymbol{a}= 10$  |
> | ---           |  ---  | --- | --- | --- | --- | --- |
> | PT-MAP        |  52.9 | 62.6| 73.8| 77.7| 80.3| 82.2|
> | BDCSPN        | 81.3  | 83.4| 85.3| 86.1| 85.3| 85.3|
> | LaplacianShot | 87.3  | 86.3| 84.5| 84.0| 84.6| 84.5|
> | LR+ICI        | 80.0  | 83.3| 86.4| 87.1| 87.5| 87.8|
> | TIM           | 73.9  | 81.0| 85.3| 86.8| 86.5| 86.7|
> | $\alpha$-TIM  | 88.2  | 87.9| 86.5| 85.5| 85.6| 85.2|
>
> #### CUB
> | Method        | $\boldsymbol{a}= 0.5$ |  $\boldsymbol{a}= 1$ |   $\boldsymbol{a}= 3$ |   $\boldsymbol{a}= 5$ |   $\boldsymbol{a}= 7$ | $\boldsymbol{a}= 10$  |
> | ---           |  ---  | --- | --- | --- | --- | --- |
> | PT-MAP        |  54.4 | 62.9| 74.9| 79.6| 81.6| 83.7|
> | BDCSPN        | 82.1  | 85.7| 88.0| 88.5| 88.4| 88.7|
> | LaplacianShot | 88.5  | 87.5| 86.8| 86.8| 86.5| 86.2|
> | TIM           | 77.2  | 83.3| 88.0| 88.8| 89.6| 89.8|
> | $\alpha$-TIM  | 90.5  | 90.6| 89.5| 89.5| 89.0| 88.9|
>
> ## Table B: Additional meta-learning methods on ***mini***-ImageNet with RN-18
>
> | Method        | 1-shot |  5-shot | 10-shot | 20-shot |
> | ---           | ---    | ---     |  ---    | ---     |
> | MAML          | 47.6 (&#8595;3.8) | 64.5 (&#8595;5.0) | 66.2 (&#8595;5.7) | 67.2 (&#8595;3.6) |
> | Versa         | 47.8 (&#8595;2.2) | 61.9 (&#8595;3.7) | 65.6 (&#8595;3.6) | 67.3 (&#8595;4.0) |

---

> ### Author Response · Authors · 2021-08-10
> **On the imbalance setting**
>
> 1- *The authors use one $\boldsymbol{a}$ value for the Dirichlet distribution (lines 257-264) to model query set imbalance during evaluation. It might be appropriate to evaluate several distributions or vary the $\boldsymbol{a}$ in the Dirichlet distribution.*
>
>
> In Fig. A (anonymous link provided) and Tables A and B (comment below), we provide additional results over the three benchmarks for different values of Dirichlet's
> parameter $\boldsymbol{a}$ for modeling the query set during testing, including $\boldsymbol{a}= 1\cdot\mathbb{1}_K$ (distributions sampled
> uniformly at random), $\boldsymbol{a}= 0.5\cdot\mathbb{1}_K$ (extreme imbalanced-distribution sampling, cf Fig. B) and different levels of class
> balance ($\boldsymbol{a}= 3\cdot\mathbb{1}_K$, $\boldsymbol{a}= 5\cdot\mathbb{1}_K$, $\boldsymbol{a}= 7\cdot\mathbb{1}_K$ and
> $\boldsymbol{a}= 10\cdot\mathbb{1}_K$). In fact, for uniformly-random sampling ($\boldsymbol{a}= 1\cdot\mathbb{1}_K$)
> and imbalanced-distribution sampling ($\boldsymbol{a}= 0.5\cdot\mathbb{1}_K$), the performance of $\alpha$-TIM is even higher
> and the performances of several existing methods is more severely affected (by important margins), which further strengthen the message of our
> work (Thanks for bringing this!). From the plots, we can observe that $\alpha$-TIM outperforms significantly the other methods when we tend to
> uniformly-random testing tasks ($\boldsymbol{a}= 1\cdot\mathbb{1}_K$), while remaining competitive/stable when we tend to balanced testing tasks (i.e., the Dirichlet parameter is increasing). These results illustrate the flexibility of $\alpha$-divergences, and are in line with the technical
> analysis we provided in section 5.2 and Fig. 2.
>
>
> 2- *It is not clear why this distribution ($\boldsymbol{a}= 2.\cdot\mathbb{1}_K$) was picked over others.*
>
> As for the choice we made in the paper, while the validation tasks are also uniformly-random ($\boldsymbol{a}= 1.\cdot\mathbb{1}_K$), our rationale behind choosing $\boldsymbol{a}= 2\cdot\mathbb{1}_K$ for the testing tasks was to reflect the fact that extremely imbalanced tasks (i.e., only one class is present in the task) are less likely to happen in practical scenarios ( $\boldsymbol{a}= 2\cdot\mathbb{1}_K$ is illustrated in Fig. A, left).
> However, we do fully agree with the reviewer that providing results for various values of $\boldsymbol{a}$ would be much more informative for readers. In particular, uniformly-random sampling ($\boldsymbol{a}= 1\cdot\mathbb{1}_K$), as in the prior works mentioned by the reviewer, is important to provide to readers. Generating imbalanced distributions is also informative, although one may argue that we are replacing a perfect-balance artefact with an imbalance artefact. We will add Fig. A (performance versus $\boldsymbol{a}$) to the paper, and the results for uniformly-random sampling to the main Table.
>
>
> 3- *The class imbalance community uses a range of distributions, for example, step imbalance and linear imbalance (Buda et al., 2018), long-tail imbalance (Wang et al., 2018), and in few-shot previous work sampled uniformly at random [32,33,34,35], or a combination of these distributions [35].*
>
> Finally, we would like to mention that, initially in the project, we considered the specific linear imbalance
> mentioned by the reviewer and the results/conclusions are quite consistent with Dirichlet's sampling
> with $\boldsymbol{a}=2\cdot\mathbb{1}_K$. However, we abandoned this in favor of introducing a more general Dirichlet sampling because it could
> also covers these settings and we did not want to replace a perfect-balance artefact with a specific imbalance artefact. As illustrated in Fig. C, both linear and step imbalance correspond to particular spots within the simplex, which are also covered by $\boldsymbol{a}=2\cdot\mathbb{1}_K$. We believe introducing more principled Dirichlet sampling to the few-shot
> literature is also an important contribution of our work, and which might be even very useful to the class-imbalance community.

---

> ### Author Response · Authors · 2021-08-10
> **Preamble**
>
> We thank the reviewer for his review and address his concerns point by point in what follows, including the requested additional results. Please note that figures indexed with a letter (e.g Fig. A) can be found at the following anonymous link: https://github.com/anonymous6496/rebuttal, while figures indexed with a number are references to figures of the main paper.

---

> ### Comment · Area_Chair_55m7 · 2021-08-30
> **[URGENT] Your thoughts?**
>
> Dear reviewer,
>
> The authors provided a thorough response to your review. What do you think of it? Other reviewers agree on acceptance; do you agree?
>
> Best regards,
>
> Your AC

---

> > ### Comment · Reviewer_maGM · 2021-08-31
> > **Increased my score**
> >
> > I thank the AC for the prompt - I have increased my scores.

---

### Official Review · Reviewer_KiFQ · 2021-07-17

**Rating:** 7
**Confidence:** 4

**Summary:**

The researchers use transductive inference and introduce the effect of arbitrary class distributions within the query sets of few-shot tasks, rather than using class-balanced tasks, by removing the class-balance artefact. They do this by modeling the marginal probabilities of the
classes as Dirichlet-distributed random variables, rather than from a known and fixed uniform distribution.
They assess their model by comparing transductive methods over 3 datasets, and show their experiment setting’s performance drops compared to inductive methods. They also propose a generalization of the mutual-information loss based on alpha-divergences, which is an extension of the Shannon mutual information and tolerates class-distribution variations more effectively.

**Limitations And Societal Impact:**

The authors have not addressed the limitations of their work.

**Main Review:**

The quality of the paper was fairly good. It is significant as it builds upon the current few-shots learning landscape, which utilizes the transductive and inductive methods. They propose the Dirichlet distribution to sample query sets as it creates a realistic setting, instead of using the current artificially balanced uniform distribution setting,

The paper illustrates that transductive few-shot methods claiming significant gains over inductive tasks may perform worse when evaluated with their realistic setting, based on their results. The paper demonstrates the conclusion with thorough experiments.

The paper also proposes alpha TIM formulation, with better generalizability n when training a few-shot scenario and sheds light on the use of alpha-divergences in such applications. The experiments also demonstrate that the alpha TIM consistently performs better with the same budget of hyper-parameter optimization as the standard TIM.

The hardware requirement is low, requiring only a single 1080 Ti GPU.

Furthermore, in terms of the clarity of the paper, it flowed well. The paper should be reproducible.
Typo on the caption of Table 1: "Comparaisons"

**Time Spent Reviewing:**

3 hours

---

> ### Author Response · Authors · 2021-08-10
> **Acknowledgment**
>
> We thank the reviewer for the positive comments on the significance of our results, on the thorough experiments, and on the novelty of both our $\alpha$-divergence-based method and Dirichlet-based sampling.

---

### Official Review · Reviewer_3EcU · 2021-07-22

**Rating:** 7
**Confidence:** 4

**Summary:**

This paper studies transductive evaluation in few-shot learning, which allows the model to use statistics of all the unlabeled query set examples when making class predictions. Models evaluated in transductive fashion typically attain higher metrics by using this extra information. Most work on transductive evaluation, however, assumes that the query set is evenly distributed across the possible classes. This is of course an unrealistic assumption for the real world and this paper specifically studies this aspect: (1) They evaluate in a more realistic transductive evaluation scenario where the class counts within the query set are sampled according to a Dirichlet distribution and show that most transductive evaluation-based methods perform worse in this setup; (2) They propose a modification of an existing model for transductive evaluation to work better in this scenario where the class counts can be varied in the query set.

**Ethical Concerns:**

No ethical concerns.

**Limitations And Societal Impact:**

Though not discussed, the societal impact of this work is probably similar to that of any other few-shot method.

**Main Review:**

Originality: the paper considers an under-studied but important aspect of transductive evaluation that to my knowledge hasn't been considered in previous work. Additionally the authors propose a modification to an existing transductive evaluation method [1] to deal with varied class makeup in the query set. The extension is simple but well-motivated as there was careful analysis to show why it would be expected to work better.

Quality: the submission seems very technically sound. The proposal for their extension is backed up by strong reasoning and the experiments seem to be conducted well.

Clarity: the paper was very well-written. It motivates the problem and why it is important well. The setup of the new sampling scheme for query set class distribution was also described well. Lastly, the description of previous methods for transductive shot-learning and the proposed extension were also stated clearly and were easy to follow.

Significance: the paper presents some interesting results: (1) It shows how most of the transductive evaluation papers are benefitting from the uniform class distribution assumption in the query set. In fact, the authors observe that more than half of the transductive methods studied perform worse than inductive baselines in this more realistic setting. This definitely brings to question some of the value of this type of work as they are taking advantage of an artificial scenario. (2) The proposed transductive method has strong performance, as its the leading method for mini-ImageNet and tiered-ImageNet in most cases and sometimes gives benefit up to 2-3% compared to the leading transductive method.

[1] Boudiaf et al. Transductive information maximization for few-shot learning. NeurIPS 2020.

**Time Spent Reviewing:**

3

---

> ### Author Response · Authors · 2021-08-10
> **Acknowledgment**
>
> We greatly appreciate the strong support to our work, and the positive comments on the significance of the results in the context of transductive few-shot learning and on the novelty/performance of the proposed $\alpha$-divergence-based method.

---

### Author Response · Authors · 2021-08-10
**Summary (to all reviewers)**

We thank all the reviewers for the insightful/constructive comments, and we are pleased that three reviewers pointed to the significance of our new setting in the context of transductive few-shot learning, and to the novelty of our α-divergence method and Dirichlet sampling. The borderline scores by reviewers maGM and W251 are stemming from requests of additional results/evaluations, which we provide below in this rebuttal.

---

### Decision · Program_Chairs · 2021-09-27

**Decision:**

Accept (Poster)

**Comment:**

The submission investigates the effect of a non-uniform marginal label distribution on transductive approaches to few-shot classification. When the marginal label distribution of test episodes is sampled from a Dirichlet distribution, the paper finds that state-of-the-art transductive approaches suffer a substantial performance drop. The submission also introduces a generalization of the mutual information loss which better handles class imbalance and is shown to outperform competing approaches in that setting.

Reviewers found that the paper is well-written and sheds light on an underexplored and important aspect of evaluating transductive approaches to few-shot classification. Overall they found that the proposed generalization of the mutual information loss is technically sound and convincingly backed up by experiments.

Some reviewers expressed concerns about the experimental design (the value of alpha and the number of values tried, its optimal value as a function of the shot setting, the choice of baselines to compare against) which were addressed to their satisfaction by the authors.

Ultimately, all reviewers agree that the paper should be accepted. I therefore recommend acceptance.